# Adaptive Transductive Inference via Sequential Experimental Design with Contextual Retention

**Tareq Si Salem**
Huawei Technologies, Paris Research Center, France
`tareq.si.salem@huawei.com`

## Abstract

This paper presents a three-stage framework for active learning, encompassing data collection, model retraining, and deployment phases. The framework's primary objective is to optimize data acquisition, data freshness, and model selection methodologies. To achieve this, we propose an online policy with performance guarantees, ensuring optimal performance in dynamic environments. Our approach integrates principles of sequential optimal experimental design and online learning. Empirical evaluations validate the efficacy of our proposed method in comparison to existing baselines.

## 1 Introduction

The development of Machine Learning (ML) models is intrinsically reliant on the availability of data. Data plays a critical role in various stages of the ML model lifecycle, including parameter optimization, evaluation of inferential capabilities, and potential refinements to the model architecture. However, the acquisition of suitable data frequently constitutes a significant bottleneck in the training pipeline. The process of obtaining relevant data or measurements can be both costly and time-intensive and is often subject to resource constraints. These constraints may manifest in diverse forms, including restrictions on sample size, temporal constraints, or computational limitations. The challenge is further compounded when labels for these datasets are unavailable and are costly to acquire (e.g., clinical trials [42], drug discovery [5]). In such scenarios, strategic decisions regarding data collection or experimental design become essential. The selection of the most informative samples for a given task is known as *optimal experimental design* (OED) [40, 10] within the field of statistics. The primary objective in OED is to maximize information gain about an unknown model within the confines of a limited budget. OED has long been an essential part of statistical modeling, from the design of clinical trials [42, 14, 46], medical imaging [39], materials science [19], biological process models [41, 18], networked systems [32], bandits [17, 31], and regression problems in general [16, 27, 49, 21]. For a comprehensive overview of OED methodologies and applications, the reader is directed to the following surveys [47, 40, 25].

In many real-world applications of ML, the performance of a model remain unknown until it is deployed and interacts with its operational environment. This inherent uncertainty necessitates an iterative approach to experimental design, where subsequent experiments are informed by the outcomes of previous ones. This adaptive methodology, known as sequential optimal experimental design (SOED), aims to effectively mitigate uncertainty by dynamically adjusting the experimental design based on accumulating knowledge. SOED presents a unique challenge in that the optimal design at any given stage depends on the anticipated sequence of future predictions. A common suboptimal approach is to employ a greedy strategy, utilizing historical data to determine the seemingly best experiment at each step without explicitly considering the long-term consequences of this decision [22, 30, 15, 9]. This myopic approach fails to account for the potential impact of

Workshop on Bayesian Decision-making and Uncertainty, 38th Conference on Neural Information Processing Systems (NeurIPS 2024).

early design choices on the overall optimization process. Recently, online learning has emerged as a promising framework for addressing SOED by incorporating feedback mechanisms into the design process [24, 26, 45]. By learning from past experiences and adapting its strategy accordingly, online learning offers a more sophisticated approach to sequential decision-making in experimental design. Furthermore, the deployment of machine learning models often encounters the challenge of concept drift, which refers to the dynamic nature of the relationship between input features and the target variable [20, 34, 43, 50, 36]. Significant disruptions, such as the COVID-19 pandemic, can precipitate substantial alterations in traffic patterns. These alterations subsequently introduce significant concept drift in traffic forecasting methodologies, thereby challenging the efficacy of predictive models [33, 35]. Consequently, this necessitates the development of training strategies that effectively tune data freshness and recency to maintain predictive accuracy. Traditional approaches, predominantly focused on variance minimization through experimental design, are insufficient in addressing the bias introduced by concept drift. Consequently, a crucial question arises:

*How can we effectively optimize data acquisition, data freshness, and model selection methodologies in dynamic environments characterized by concept drifts?*

To address this challenge, this work introduces a novel active learning framework. This framework operates in three distinct stages: data collection, model retraining, and deployment. In the initial data collection phase, a policy guides the selection of informative experiments from a predefined pool of potential candidates. This selection process aims to maximize the value of acquired labels, thereby enhancing learning efficiency. The policy can request labels for multiple experiments concurrently, subject to a constraint on the number of simultaneous queries. These labeled datasets are then stored in a local repository with a fixed capacity, utilizing an eviction strategy to manage storage limitations. Periodically, the accumulated labeled data is used to retrain a machine learning model. This model, updated with an appropriate selection of fresh dataset from the local repository, is then deployed to predict labels for new, incoming queries. Unlike traditional inductive learning approaches that focus solely on the initial pool of experiments, this framework adopts a transductive learning perspective [11, 48, 6, 51, 12]. This means that the policy's experimental design choices are optimized not only with respect to the initial pool but also in relation to the sequence of revealed queries. The policy receives feedback on its predictions in the form of noisy prediction errors, allowing it to adapt and refine its strategy over time. The ultimate goal of the policy is to minimize its regret, which quantifies the performance difference between the policy's predictions and those of an optimal policy possessing complete information about the underlying data generating process. This minimization of regret ensures that the active learning framework efficiently learns and adapts to the underlying phenomenon, even in the presence of noise and uncertainty.

The remainder of this paper is organized as follows. Section 2 presents the problem formulation. A theoretical analysis of the problem is conducted in Section 3. Finally, the effectiveness of the proposed approach is numerically demonstrated in Section 4.

## 2 Problem Formulation

### 2.1 System Model

The overall system model is illustrated in Figure 1. A list of the notation employed throughout this paper can be found in the Appendix.

**Data Collection.** The policy has access to a pool of experiments $\mathcal{X} \subset \mathbb{R}^d$ to collect labels from a variety of experimental sources, such as sensors, surveys, and databases. A data retention policy is implemented, periodically purging datasets that exceed a predetermined age threshold $\tau \in \mathbb{N}$. This practice adheres to data privacy regulations (e.g., GDPR [13], CCPA/CPRA[1, 38]). At each time slot $t$, the policy is allocated a fixed experimental budget of $M \in \mathbb{N}$ experiments. The set of all feasible experimental designs is defined as

$$\triangle_{\mathcal{X}} \triangleq \left\{ \boldsymbol{\pi} \in [0,1]^{\mathcal{X}} : \|\boldsymbol{\pi}\|_1 = 1 \right\}.$$

For a given continuous design $\boldsymbol{\pi} \in \triangle_{\mathcal{X}}$, the learner allocates $M\pi_{\boldsymbol{x}} \in [0, M]$ experiments to type $\boldsymbol{x} \in \mathcal{X}$. At time $t$, the system acquires a dataset $\mathcal{D}_t$ of $M$ experiment-labels pairs in $\mathcal{X} \times \mathbb{R}$, following a design $\boldsymbol{\pi}_t \in \triangle_{\mathcal{X}}$. A rolling window of $\tau + 1$ datasets is maintained, discarding older ones. The labels $y$ are related to the experiment $\boldsymbol{x}$ according to a noisy linear model $y = \boldsymbol{x} \cdot \boldsymbol{\beta}_t^\star + n$, where $n \sim \mathcal{N}(0, \sigma^2)$ is a Gaussian noise and $\boldsymbol{\beta}_t^\star$ is the *time-varying* true model.

**Model Retraining.** To address the challenge of concept drift during the model updating phase, we introduce a data-freshness parameter, denoted as $\tau_t \in \mathcal{T} \triangleq \{0, 1, \ldots, \tau\}$. This parameter dictates the recency of the data used for model training. Specifically, at each time slot $t$, we employ a least-squares estimator (LSE), denoted as $\hat{\boldsymbol{\beta}}_t$, to generate future predictions. The LSE model is trained exclusively on the $(\tau_t + 1)$ most recent data points avail-

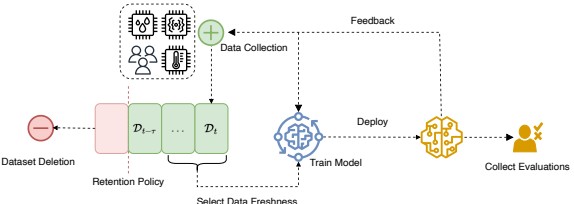

Figure 1: System Model

able at time $t$. The model is given by $\hat{\boldsymbol{\beta}}_t = \frac{1}{M} \left(\boldsymbol{X}^\intercal \mathrm{diag}\left(\boldsymbol{\pi}_{t-\tau_t:t}\right) \boldsymbol{X}\right)^{-1} \boldsymbol{X}^\intercal \boldsymbol{y}\left(\mathcal{D}_{t-\tau_t:t}\right)$, where $\boldsymbol{X} = (\boldsymbol{x}^\intercal)_{\boldsymbol{x} \in \mathcal{X}} \in \mathbb{R}^{\mathcal{X} \times d}$ is the experiments' matrix, and $\boldsymbol{y}(\mathcal{D}) = \left(\sum_{(\boldsymbol{x}, y_i) \in \mathcal{D}_{\boldsymbol{x}}} y_i\right)_{\boldsymbol{x} \in \mathcal{X}} \in \mathbb{R}^{\mathcal{X}}$ is the labels' vector.

**Model Deployment.** During deployment, the trained model $\hat{\boldsymbol{\beta}}_t$ is used to predict labels for experiments $\boldsymbol{x}_t \in \mathcal{Z} \subset \mathbb{R}^d$. User feedback in the form of prediction errors is collected to refine the model. Specifically, the squared error $\xi_t = (y_t - \hat{y}_t)^2$ is provided, where $\hat{y}_t = \boldsymbol{x}_t \cdot \hat{\boldsymbol{\beta}}_t$ is the predicted label and $y_t = \boldsymbol{x}_t \cdot \boldsymbol{\beta}_t^\star + n$ is the label with noise $n \sim \mathcal{N}(0, \sigma^2)$. This feedback signal informs the learner about the model's performance and guides future data collection to improve accuracy. Both the feedback signal $\xi_t$ and the corresponding query $\boldsymbol{x}_t$ are used to guide future model selection.

## 2.2 Policies and Performance Metric

In this section, we provide a formal description of the policy that governs the collection of data through experimental designs and the freshness of data used to retrain the most recent model.

**Prediction Error and Bias–Variance Tradeoff.** The accuracy of the model $\hat{\boldsymbol{\beta}}_t$ for a query point $\boldsymbol{x}_t \in \mathcal{Z}$ with noisy label $y_t = \boldsymbol{x}_t \cdot \boldsymbol{\beta}_t^\star + n \sim \mathcal{N}(\boldsymbol{x}_t \cdot \boldsymbol{\beta}_t^\star, \sigma^2)$ is measured by its expected prediction error (EPE). This EPE depends on the experimental designs $(\boldsymbol{\pi}_{t-\tau}, \ldots, \boldsymbol{\pi}_t)$ and a data-freshness parameter $\tau_t \in \mathcal{T}$ controlling the influence of past data. The EPE is defined as follows: $f_t(\boldsymbol{\pi}_{t-\tau_t}, \ldots, \boldsymbol{\pi}_t) \triangleq \mathbb{E}[\xi_t] = \mathbb{E}[(y_t - \hat{y}_t)^2]$. In the following Proposition, we clearly delineate the contributions of experimental design selection and data-freshness selection, by decomposing the EPE on query $\boldsymbol{x}_t$ into its variance and bias components.

**Proposition 1.** *Under designs $\{\boldsymbol{\pi}_s\}_{s=t-\tau}^t \in \triangle_{\mathcal{X}}^{\tau+1}$, the EPE of the LSE on experiment $\boldsymbol{x}_t \in \mathcal{Z}$ at time $t$ under data-freshness window size $\tau + 1$ is*

$$f_t(\boldsymbol{\pi}_{t-\tau}, \ldots, \boldsymbol{\pi}_t) = \sigma^2 + \boldsymbol{x}_t^\intercal \mathrm{cov}(\hat{\boldsymbol{\beta}}_t)\boldsymbol{x}_t + \left(\boldsymbol{x}_t \cdot \left(\mathbb{E}\left[\hat{\boldsymbol{\beta}}_t\right] - \boldsymbol{\beta}_t^\star\right)\right)^2, \qquad (1)$$

*where $\mathbb{E}\left[\hat{\boldsymbol{\beta}}_t\right] = \boldsymbol{V}^{-1}(\boldsymbol{\pi}_{t-\tau_t:t}) \left(\sum_{\boldsymbol{x} \in \mathcal{X}} \boldsymbol{x}\boldsymbol{x}^\intercal \sum_{s=t-\tau}^t \pi_{s,\boldsymbol{x}} \boldsymbol{\beta}_s^\star\right)$ and $\mathrm{cov}(\hat{\boldsymbol{\beta}}_t) = \frac{\sigma^2}{M} \boldsymbol{V}^{-1}(\boldsymbol{\pi}_{t-\tau_t:t})$.*

The proof is provided in Appendix B.1. The bias-variance trade-off is evident in the decomposition. The expected prediction error is divided into three components: (a) irreducible variance due to noise in the labels, (b) variance related to data-freshness and experimental designs, and (c) bias reflecting model drift. In the absence of significant drift, a larger data-freshness window is beneficial. However, under significant drift, a smaller window is preferable, though increasing variance. Minimizing variance through careful experimental design is always advantageous.

**Online Policies.** The role of a policy is to select appropriately experimental designs $\boldsymbol{\pi}_t \in \triangle_{\mathcal{X}}$ and data-freshness parameter $\tau_t \in \mathcal{T}$ at every timeslot $t$, and adapts its decisions upon seeing the query $\boldsymbol{x}_t$ and feedback $\xi_t$. Formally, at timeslot $t$, the system adapts its state according to a randomized policy $\mathcal{P}_t : (\triangle_{\mathcal{X}} \times \mathcal{T} \times \mathcal{Z} \times \mathbb{R})^t \rightarrow \triangle_{\mathcal{X}} \times \mathcal{T}$, defined as $(\boldsymbol{\pi}_{t+1}, \tau_{t+1}) = \mathcal{P}_t\left(\{\boldsymbol{\pi}_s, \tau_s, \boldsymbol{x}_s, \xi_s\}_{s=1}^t\right)$.

**Performance Metric.** We compare the performance of the sequence of designs and data-freshness parameters w.r.t. the best design in hindsight and data-freshness window size after seeing all the queries in terms of the EPE. Formally,

$$\mathfrak{R}_T\left(\boldsymbol{\mathcal{P}}\right) \triangleq \mathbb{E}\left[\sum_{t=\tau+1}^T f_t(\boldsymbol{\pi}_{t-\tau_t}, \ldots, \boldsymbol{\pi}_t) - \sum_{t=\tau+1}^T f_t(\boldsymbol{\pi}_{t-\tau_t^\star}^\star, \ldots, \boldsymbol{\pi}_t^\star)\right], \qquad (2)$$

where $\boldsymbol{\pi}_t^\star$ and $\tau_t^\star$ for $t \in [T]$ are the minimizers of the aggregate expected prediction error, and the expectation is taken with respect to the randomness in the environment and the policy $\mathcal{P}$.

If the regret $\mathfrak{R}_T(\mathcal{P})$ is sublinear, then the policy asymptotically achieves on average the performance of a policy that selects the optimal experimental designs and data-freshness parameters in hindsight.

## 3 Theoretical Analysis

Our theoretical analysis reveals that the optimization of experimental design and data-freshness can be efficiently decoupled into two interrelated subproblems. We propose a policy utilizing Online Mirror Descent (OMD) [8, 44, 23] to simultaneously address these subproblems.

### 3.1 Decoupling of Experimental Design and Data-freshness Decisions

We propose a decoupled approach to experimental design and data-freshness parameter selection. Firstly, we employ a variance reduction policy within the full-information online learning framework [23, 37] to determine the optimal experimental design. Secondly, we formulate a multi-armed bandit problem [31] to select the data-freshness parameters, thereby mitigating bias.

**Variance Reduction Policy.** The variance reduction policy selects a new design at time $t$ according to a mapping $\mathcal{P}_t^v$ that maps the past experimental designs $\{\boldsymbol{\pi}_s\}_{s=1}^t \in \triangle_{\mathcal{X}}^t$, and past experiment queries $\{\boldsymbol{x}_s\}_{s=1}^t \in \mathcal{Z}^t$ to a new design $\boldsymbol{\pi}_{t+1}$ given by $\boldsymbol{\pi}_{t+1} = \mathcal{P}_t^v(\{\boldsymbol{\pi}_s, \boldsymbol{x}_s\}_{s=1}^t)$. The policy incurs costs in the form of the $\boldsymbol{x}_t$-optimal design objective in Definition 2, and has the following regret: $\mathfrak{R}_T^v(\mathcal{P}^v) \triangleq \sum_{t=1}^T v_t(\boldsymbol{\pi}_t) - \min_{\{\boldsymbol{\pi}_t^\star\}_{t=1}^T \in \triangle_{\mathcal{X}}^T} \sum_{t=1}^T v_t(\boldsymbol{\pi}_t^\star)$. Note that the cost $v_t$ are fully determined once $\boldsymbol{x}_t$ is made available.

**Bias Reduction Policy.** The bias reduction policy operates on top of the variance reduction policy. In particular, at timeslot $t$, the data-freshness parameter $\tau_{t+1}$ is the output of the mapping $\mathcal{P}_t^b$ that maps the past data-freshness parameters $\{\tau_s\}_{s=1}^t \in \mathcal{T}^t$ and prediction error feedback $\{\xi_s\}_{s=1}^t \in \mathbb{R}^t$ according to the mapping $\tau_{t+1} = \mathcal{P}_t^b(\{\tau_s, \xi_s\}_{s=1}^t)$. Note that the coupling between the variance reduction policy and bias reduction policy is implicitly encoded in the prediction error $\xi_t$ as this error depends on both decisions. The policy incurs the prediction errors, and has the following regret: $\mathfrak{R}_T^b(\mathcal{P}^b) \triangleq \mathbb{E}\left[\sum_{t=\tau+1}^T \xi_t - \min_{\{\tau_t^\star\}_{t=1}^T \in \mathcal{T}^T} \sum_{t=\tau_t^\star+1}^T f_t(\boldsymbol{\pi}_{t-\tau_t^\star}, \ldots, \boldsymbol{\pi}_t)\right]$. The setup of bias reduction policy corresponds to a non-stationary setup of the multi-armed bandit problem where $\mathcal{T}$ is the set of the arms [31, 7].

**VBR Policy.** The variance and bias reduction (VBR) policy denoted by $\mathcal{P}^{v+b}$ is the policy that determines experimental designs $\boldsymbol{\pi}_t$ according to the variance reduction policy $\mathcal{P}^v$ and the data-freshness parameter $\tau_t$ according to the bias reduction policy $\mathcal{P}^b$ for any $t \in [T]$. Formally, at timeslot $t$, the policy is given by the mapping $\mathcal{P}_t^{v+b} : (\triangle_{\mathcal{X}} \times \mathcal{T} \times \mathcal{Z} \times \mathbb{R})^t \to \triangle_{\mathcal{X}} \times \mathcal{T}$ given by $\mathcal{P}_t^{v+b} \triangleq (\mathcal{P}_t^v, \mathcal{P}_t^b)$. The variance and bias reduction policy enjoys the following regret guarantee:

**Theorem 1.** *Under Assumptions 1–3, let $\{\boldsymbol{x}_t\}_{t=1}^T \in \mathcal{Z}^T$ be the sequence of queries, $\{\boldsymbol{\pi}_t^\star\}_{t=1}^T \in \triangle_{\mathcal{X}}^T$ be the sequence of optimal experimental designs and $\{\tau_t^\star\}_{t=1}^T \in \mathcal{T}^T$ is the sequence of data-freshness windows. The regret (2) of the variance and bias reduction policy $\mathcal{P}^{v+b}$ satisfies:*

$$\mathfrak{R}_T(\mathcal{P}^{v+b}) = \mathfrak{R}_T^b(\mathcal{P}^b) + \mathfrak{R}_T^v(\mathcal{P}^v) + \mathcal{O}\left(P_T^v + P_T^{\star,v}\right), \tag{3}$$

*where $P_T^{\star,v} = \sum_{t=1}^T \left\|\boldsymbol{\pi}_t^\star - \boldsymbol{\pi}_{t+1}^\star\right\|_1$ and $P_T^v = \sum_{t=1}^T \left\|\boldsymbol{\pi}_t - \boldsymbol{\pi}_{t+1}\right\|_1$ are the path-lengths of the OEDs and variance reduction policy.*

The proof is provided in Appendix E. In the next section, we provide a specific instantiations of the variance reduction and bias reduction policies.

### 3.2 Entropic-VBR Policy

We introduce the Entropic-VBR policy (Appendix C.4), which achieves sublinear regret. Our approach provides a unified treatment of both full-information and bandit settings. To overcome the challenges of a non-stationary environment and bias reduction in the bandit setting, we meticulously

instantiate the OMD framework. This involves carefully constructing gradient estimates and selecting an appropriate mirror map to ensure simultaneous regret guarantees. Leveraging the results of Corollary 2, Theorem 4, and Theorem 1, we establish a comprehensive regret guarantee for the Entropic-VBR policy. Formally,

**Corollary 1.** *Under Assumptions 1–3, let $\{\boldsymbol{x}_t\}_{t=1}^{T} \in \mathcal{Z}^T$ be the sequence of queries, $\{\boldsymbol{\pi}_t^{\star}\}_{t=1}^{T} \in \triangle_{\mathcal{X},\sigma}^{T}$ be the sequence of optimal experimental designs and $\{\boldsymbol{p}_t^{\star}\}_{t=1}^{T} \in \triangle_{\mathcal{T}}^{T}$ is the sequence of comparator data-freshness windows, with path lengths $P_T^{\star,v}$ and $P_T^{\star,b}$, respectively. The Entropic-VBR Policy (Appendix C.4) configured with learning rates $\eta_{\mathcal{X}} = \Theta\left(\log(1/\sigma)P_T^{\star,v}T^{-1}\right)$ and $\eta_{\mathcal{T}} = \Theta\left(\log(1/\sigma')P_T^{\star,b}T^{-1}\right)$ and $\sigma' = \Theta\left(T^{-1}\right)$ achieves the following regret:*

$$\mathfrak{R}_T(\boldsymbol{\mathcal{P}}^{\text{Entropic}-\text{VBR}}) = \mathcal{O}\left(\sqrt{\log(1/\sigma)P_T^{\star,v}T} + \sqrt{\log(T)P_T^{\star,b}T} + P_T^{\star,v}\right). \tag{4}$$

The sequences in $\triangle_{\mathcal{X},\sigma}$ are $(1+\sigma)$-competitive w.r.t. sequences in $\triangle_{\mathcal{X}}$ (Prop. 2 in the Appendix).

## 4  Numerical Experiments

**Experimental Setup.** To evaluate the performance of our proposed methodology, we constructed a synthetic experimental setting, as illustrated in Figure 2 (a). Full description is in Appendix F.

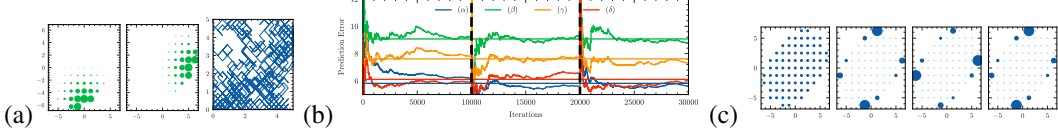

(a)          (b)          (c)

Figure 2: Subfig. (a): Query distributions, and true model drift ($10^3$ initial iterations). Subfig. (b): Time-avged prediction error for three intervals: $0 \le t \le \frac{1}{3}T$, $\frac{1}{3}T < t \le \frac{2}{3}T$, and $\frac{2}{3}T < t \le T$. Subfig. (c): Initial and selected designs at $t = \frac{1}{3}T$, $t = \frac{2}{3}T$, and $t = T$.

**Discussion.** Our evaluation in Figure 2 (b) shows that a uniform experimental design with the maximum freshness window is the least effective baseline (indexed as $\beta$). Optimizing the data-freshness window improves performance (indexed as $\gamma$), and further gains are achieved by optimizing the experimental design (indexed as $\delta$). Our proposed policy (indexed as $\alpha$) outperforms the baseline, demonstrating its ability to identify optimal sequences of designs and freshness windows. Figure 2 (c) represents how the policy adapts to evolving query distributions. The temporal evolution of the learned distribution over window sizes in Figure 3 (Appendix) reveals that the optimal window size under these conditions is not immediately apparent.

## 5  Conclusion

This work introduced a novel framework that explicitly accounts for the evolving relationship between data freshness and model performance, encompassing data collection, data freshness decisions, and model retraining within a limited-capacity cache. A rigorous theoretical analysis revealed the inherent variance-bias trade-off and motivated a decoupled approach to address this challenge. This approach involved leveraging OCO for variance reduction in experimental design and formulating a non-stationary MAB problem for bias mitigation through data freshness parameter selection.

As avenues for future work, extending the inference model to encompass non-linear relationships is of considerable interest. Reproducing kernel methods [3, 4] present a promising initial direction for leveraging the proposed framework, as they permit analogous derivations. Additionally, exploring more general noise models beyond the Gaussian noise considered herein would enhance the framework's applicability.

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

# Technical Appendix

## A  Formal Assumptions and Definitions

We provide a summary of the notation used in this document.

Table 1: Notation summary

| | **Notational Conventions** |
|---|---|
| $[n]$ | Set $\{1, 2, \ldots, n\}$ |
| $S^V$ | Set of functions from set $V$ to set $S$ |
| $x, \boldsymbol{x}, \boldsymbol{X}$ | Scalar, column vector, matrix |
| $\|\boldsymbol{x}\|_{\boldsymbol{A}}$ | Norm $\|\boldsymbol{x}\|_{\boldsymbol{A}} = \sqrt{\boldsymbol{x}^\intercal \boldsymbol{A} \boldsymbol{x}}$ for p.d. matrix $\boldsymbol{A}$ |
| $\boldsymbol{x}_{t':t}$ | Summation $\sum_{s=t'}^{t} \boldsymbol{x}_s$ |
| $\mathcal{D}_{t':t}$ | Union of sets $\bigcup_{s \in \{t', t'+1, \ldots, t\}} \mathcal{D}_s$ |
| $\boldsymbol{A} \succ (\succeq) 0$ | Positive (semi-)definite matrix satisfying $\boldsymbol{x}^\intercal \boldsymbol{A} \boldsymbol{x} > (\geq) 0$ for $\boldsymbol{x} \in \mathbb{R}^d \setminus \{\boldsymbol{0}\}$ |
| | **Experimental Design** |
| $\mathcal{X}$ | Set of experiments $\mathcal{X} \subset \mathbb{R}^d$ |
| $\mathcal{Z}$ | Set of test experiments $\mathcal{Z} \subseteq \mathbb{R}^d$ |
| $\boldsymbol{\beta}_t^\star$ | True model at time $t$ |
| $\hat{\boldsymbol{\beta}}_t$ | Estimation of true model at time $t$ |
| $\triangle_{\mathcal{X}}$ | Experimental design space $\triangle_{\mathcal{X}} = \left\{ \boldsymbol{\pi} \in [0,1]^{\mathcal{X}} : \|\boldsymbol{\pi}\|_1 = 1 \right\}$ |
| $\triangle_{\mathcal{X}, \sigma}$ | Restricted experimental design space $\triangle_{\mathcal{X}, \sigma} = \left\{ \frac{\boldsymbol{\pi} + \frac{\sigma}{|\mathcal{X}|}}{1+\sigma} : \boldsymbol{\pi} \in \triangle_{\mathcal{X}} \right\}$ |
| $\boldsymbol{\pi}$ | Experiemental design |
| $M$ | Number of conducted experiments under a selected design |
| $(\boldsymbol{x}, y)$ | Experiment $\boldsymbol{x} \in \mathcal{X}$ and label $y \in \mathbb{R}$ pair |
| $\mathcal{D}$ | Dataset $\mathcal{D} = \{(x_i, y_i) : i \in [M]\}$ |
| $\mathcal{D}_{\boldsymbol{x}}$ | Dataset $\mathcal{D}_{\boldsymbol{x}} = \{(x', y) \in \mathcal{D} : \boldsymbol{x}' = \boldsymbol{x}\}$ |

### A.1  Technical Assumptions

We impose the following technical assumptions for our theoretical analysis.

**Assumption 1.** *(Compact Experiments and Query Sets) Experiments $\boldsymbol{x} \in \mathcal{X}$ and $\boldsymbol{x}' \in \mathcal{Z}$ are uniformly bounded under the $\ell_2$ norm by $D_{\mathcal{X}}$ and $D_{\mathcal{Z}}$, respectively. Formally, $\|\boldsymbol{x}\|_2 \leq D_{\mathcal{X}}, \|\boldsymbol{x}'\|_2 \leq D_{\mathcal{Z}}$ for all $\boldsymbol{x} \in \mathcal{X}, \boldsymbol{x}' \in \mathcal{Z}$.*

**Assumption 2.** *(Compact Parameter Set) We assume that the true model parameters $\boldsymbol{\beta}_t^\star$ for $t \in [T]$, are uniformly bounded. Specifically, there exists a positive constant $B^\star$ such that $\|\boldsymbol{\beta}_t^\star\|_2 \leq B^\star$ for all $t \in [T]$.*

**Assumption 3.** *(Invertible Design Matrices) The matrix $\sum_{\boldsymbol{x} \in \mathcal{X}} \boldsymbol{x} \boldsymbol{x}^\intercal$ is non-singular, meaning that there exists a positive constant $\omega \in \mathbb{R}_{>0}$ such that the following inequality holds: $\sum_{\boldsymbol{x} \in \mathcal{X}} \boldsymbol{x} \boldsymbol{x}^\intercal \succeq |\mathcal{X}| \omega \boldsymbol{I} \succ 0$.*

Assumptions 1–2 guarantee the compactness of the experimental design space, the query space, and the model space. This compactness assumption is frequently employed in the analysis of learning problems [23, 44], facilitating the establishment of various theoretical properties. Furthermore, Assumption 3 ensures the invertibility of the covariance matrix $\boldsymbol{V}(\boldsymbol{\pi})$, enabling the well-definedness of the LSE.

### A.2  Technical Definitions

We introduce the following key concepts foundational to our theoretical analysis.

**Definition 1.** *Let $\sigma \in (0, 1]$ a positive real number. We define the $\sigma$-regularized simplex $\triangle_{\mathcal{X}, \sigma}$ as follows: $\triangle_{\mathcal{X}, \sigma} \triangleq \{T(\boldsymbol{\pi}) : \boldsymbol{\pi} \in \triangle_{\mathcal{X}}\}$, where $T$ is a map given by $T : \boldsymbol{\pi} \in \triangle_{\mathcal{X}} \to \frac{\boldsymbol{\pi} + \frac{\sigma}{|\mathcal{X}|} \boldsymbol{1}}{1+\sigma}$ and $\boldsymbol{1}$ denotes the all-ones vector in $\mathbb{R}^{\mathcal{X}}$.*

It is easy to verify that $\triangle_{\mathcal{X},\sigma} \subseteq \triangle_{\mathcal{X}}$. Remark that selecting designs in the above set implicitly introduces ridge-regularization/Bayesian prior (with parameter $\lambda$), which allows the covariance matrix $\boldsymbol{V}$ to be invertible for every design $\boldsymbol{\pi} \in \triangle_{\mathcal{X},\sigma}$. We define the following $\boldsymbol{x}_t$-optimal design objective (see the discussion in Sec. A.3) determined by the designs $\boldsymbol{\pi}_{t-\tau}, \dots, \boldsymbol{\pi}_t \in \triangle_{\mathcal{X},\sigma}$ and the experiment $\boldsymbol{x}_t \in \mathcal{Z}$ for $t \in [T]$.

**Definition 2.** *Consider Assumptions 1 and 3 hold. Consider $\boldsymbol{x}_t \in \mathcal{Z}$ a query experiment at timeslot $t$ and experimental designs $\boldsymbol{\pi}_{t-\tau}, \dots, \boldsymbol{\pi}_t \in \triangle_{\mathcal{X},\sigma}$ at timeslot $t \in [T]$. The $\boldsymbol{x}_t$-optimal design objective at time $t \in [T]$ under the LSE estimator is given by $v_t(\boldsymbol{\pi}_{t-\tau}, \dots, \boldsymbol{\pi}_t) \triangleq \boldsymbol{x}_t^\mathsf{T} \boldsymbol{V}^{-1}(\boldsymbol{\pi}_{t-\tau:t})\boldsymbol{x}_t$.*

Restricting the design space to $\triangle_{\mathcal{X},\sigma}$ from the original design space $\triangle_{\mathcal{X}}$ provides the following approximation guarantee:

**Proposition 2.** *(Allen-Zhu et al. [2, Proposition 3.6]) For any $\sigma \in (0,1)$, it holds for the $\boldsymbol{x}_t$-optimal design objective: $v_t(\boldsymbol{\pi}_\star) \leq v_t(\boldsymbol{\pi}_{\star,\sigma}) \leq (1+\sigma)v_t(\boldsymbol{\pi}_\star)$, where $\boldsymbol{\pi}^\star \in \operatorname{argmin}_{\boldsymbol{\pi} \in \triangle_{\mathcal{X}}} v_t(\boldsymbol{\pi})$ and $\boldsymbol{\pi}_{\star,\sigma} \in \operatorname{argmin}_{\boldsymbol{\pi} \in \triangle_{\mathcal{X},\sigma}} v_t(\boldsymbol{\pi})$.*

The above proposition implies that if the designs $\boldsymbol{\pi}_t \in \triangle_{\mathcal{X},\sigma}$ selected at timeslots $t \in [T]$ are competitive with the best design in $\triangle_{\mathcal{X},\sigma}$, then they are also competitive with the best design in $\triangle_{\mathcal{X}}$, penalized by the factor $(1+\sigma) \geq 1$.

### A.3 Experimental Design Landscape

Experimental design deals with the design of experiments in order to maximize the statistical efficiency of the resulting estimates. The objective is evaluated as a transformation of the covariance matrix of the LSE estimator. Popular choices [40] include: G(lobal)-optimality $v_G(\boldsymbol{V}) = \max \operatorname{diag}\left(\boldsymbol{X}\boldsymbol{V}^{-1}\boldsymbol{X}^\mathsf{T}\right)$, A(verage)-optimality $v_A(\boldsymbol{V}) = \operatorname{tr}\left(\boldsymbol{V}^{-1}\right)/|\mathcal{X}|$, D(eterminant)-optimality $v_D(\boldsymbol{V}) = (\det \boldsymbol{V})^{-1/|\mathcal{X}|}$, and $\boldsymbol{c}$-optimality $v_{\boldsymbol{c}}(\boldsymbol{V}) = \boldsymbol{c}^\mathsf{T}\boldsymbol{V}^{-1}\boldsymbol{c}$. The choice of criterion depends on the specific goals of the experiment. For example, if the goal is to reduce the maximum estimation error of the model over all possible experiments, then D-optimality or G-optimality would be a good choice.[1] If the goal is to to reduce the estimation error on average over all possible experiments then $V$-optimality is a good choice. If the goal is to estimate the outcome of a specific experiment $\boldsymbol{c}$ of interest with as much precision as possible, then $\boldsymbol{c}$-optimality would be a good choice.

## B Technical Lemmas

### B.1 Proof of Proposition 3.1

*Proof.* The EPE can be decomposed to variance and bias term in the following fashion. Given a query $\boldsymbol{x}_t$, the following equality holds:

$$\mathbb{E}\left[\left(y_t - \boldsymbol{x}_t \cdot \hat{\boldsymbol{\beta}}_t\right)^2\right] = \mathbb{E}\left[(y_t - \boldsymbol{x}_t \cdot \boldsymbol{\beta}_t^\star)^2\right] + \mathbb{E}\left[\left(\boldsymbol{x}_t \cdot \left(\boldsymbol{\beta}_t^\star - \hat{\boldsymbol{\beta}}_t^\star\right)\right)^2\right] = \sigma^2 + \left(\boldsymbol{x}_t \cdot \left(\boldsymbol{\beta}_t^\star - \mathbb{E}\left[\hat{\boldsymbol{\beta}}_t\right]\right)\right)^2$$

$$+ \mathbb{E}\left[\left(\boldsymbol{x}_t \cdot \left(\mathbb{E}\left[\hat{\boldsymbol{\beta}}_t\right] - \hat{\boldsymbol{\beta}}_t\right)\right)^2\right] = \sigma^2 + \left(\boldsymbol{x}_t \cdot \left(\mathbb{E}\left[\hat{\boldsymbol{\beta}}_t\right] - \hat{\boldsymbol{\beta}}_t\right)\right)^2 + \boldsymbol{x}_t^\mathsf{T}\operatorname{cov}(\hat{\boldsymbol{\beta}}_t)\boldsymbol{x}_t. \tag{5}$$

Applying the identity $(a+b)^2 = a^2 + b^2 + 2ab$ twice, we can further analyze the expression. The first application of this identity is valid due to the properties of the label noise, specifically $\mathbb{E}[\eta_t] = 0$ and $\mathbb{E}[\eta_t^2] = \sigma^2$. Subsequently, the second equality is derived by subtracting and adding the expected value of the estimator $\mathbb{E}[\hat{\boldsymbol{\beta}}_t]$, to the expression. Note that $\operatorname{cov}(\boldsymbol{A}\boldsymbol{y}) = \boldsymbol{A}\operatorname{cov}(\boldsymbol{y})\boldsymbol{A}^\mathsf{T}$ for some deterministic matrix $\boldsymbol{A}$. Take $\boldsymbol{A} = \boldsymbol{V}^{-1}(\boldsymbol{\pi}_{t-\tau:t})\boldsymbol{X}^\mathsf{T}$. The covariance is given by the following

$$\operatorname{cov}(\hat{\boldsymbol{\beta}}_t) = \boldsymbol{A}\operatorname{cov}(\boldsymbol{y})\boldsymbol{A}^\mathsf{T} = \boldsymbol{A}\operatorname{cov}\left(\frac{1}{M}\sum_{s=t-\tau}^t \sum_{(x,y)\in\mathcal{D}_{s,\boldsymbol{x}}} y\right)_{\boldsymbol{x}\in\mathcal{X}}\boldsymbol{A}^\mathsf{T} = \frac{\sigma^2}{M}\boldsymbol{A}\operatorname{diag}\left((\boldsymbol{\pi}_{\boldsymbol{x},t-\tau:t})_{\boldsymbol{x}\in\mathcal{X}}\right)\boldsymbol{A}^\mathsf{T} \tag{6}$$

$$= \sigma^2 \left(\boldsymbol{V}^{-1}(\boldsymbol{\pi}_{t-\tau:t})\boldsymbol{X}\right)^\mathsf{T}\operatorname{diag}\left((\boldsymbol{\pi}_{\boldsymbol{x},t-\tau:t})_{\boldsymbol{x}\in\mathcal{X}}\right)\left(\boldsymbol{V}^{-1}(\boldsymbol{\pi}_{t-\tau:t})\boldsymbol{X}^\mathsf{T}\right)^\mathsf{T} = \frac{\sigma^2}{M}\boldsymbol{V}^{-1}(\boldsymbol{\pi}_{t-\tau:t}), \tag{7}$$

---

[1]Note that D-optimality and G-optimality are interchangeable when $\mathcal{X} \subset \mathbb{R}^d$ is compact and $\operatorname{span}(\mathcal{X}) = \mathbb{R}^d$ [28].

where we used var $(a(y_1 + y_2)) = a^2$ for two i.i.d. r.v.s with variance 1 in Eq. (6), $(\boldsymbol{AB})^\mathsf{T} = \boldsymbol{B}^\mathsf{T}\boldsymbol{A}^\mathsf{T}$ and the symmetry of $\boldsymbol{V}^{-1}(\boldsymbol{\pi}_{t-\tau:t})$ in Eq. (7). Further note that we just proved $\boldsymbol{x}_t^\mathsf{T}\mathrm{cov}(\hat{\boldsymbol{\beta}}_t)\boldsymbol{x}_t = \frac{\sigma^2}{M}\|\boldsymbol{x}_t\|_{\boldsymbol{V}^{-1}(\boldsymbol{\pi}_{t-\tau:t})}$.

We have the following $\mathbb{E}\left[\hat{\boldsymbol{\beta}}_t\right] = \boldsymbol{V}^{-1}(\boldsymbol{\pi}_{t-\tau:t})\boldsymbol{X}^\mathsf{T}\mathbb{E}\left[\boldsymbol{y}\right]$. We compute the expectation of the vector of labels $\boldsymbol{y}$: $\mathbb{E}\left[\boldsymbol{y}\right] = \left(\mathbb{E}\left[\sum_{s=t-\tau}^{t}\sum_{(x,y)\in\mathcal{D}_{s,\boldsymbol{x}}}y\right]\right)_{\boldsymbol{x}\in\mathcal{X}} = \left(\sum_{s=t-\tau}^{t}\mathbb{E}\left[\sum_{(x,y)\in\mathcal{D}_{s,\boldsymbol{x}}}y\right]\right)_{\boldsymbol{x}\in\mathcal{X}} = \left(\boldsymbol{x}^\mathsf{T}\sum_{s=t-\tau}^{t}\pi_{s,x}\boldsymbol{\beta}_t^\star\right)_{\boldsymbol{x}\in\mathcal{X}}$. This concludes the proof. $\qquad\square$

**Lemma 1.** *Consider an invertible square matrix $\boldsymbol{A}(\pi) \in \mathbb{R}^{d\times d}$ parameterized by $\pi \in \mathbb{R}$. The derivative of $\boldsymbol{A}^{-1}(\pi)$ w.r.t $\pi$ is*

$$\frac{d\boldsymbol{A}^{-1}(\pi)}{d\pi} = -\boldsymbol{A}^{-1}\frac{d\boldsymbol{A}(\pi)}{d\pi}\boldsymbol{A}^{-1}. \tag{8}$$

*Proof.* Matrix $\boldsymbol{A}(\pi)$ is invertible, so $\boldsymbol{A}^{-1}(\pi)\boldsymbol{A}(\pi) = \boldsymbol{I}$. Let $A_{i,j}$ and $A_{i,j}^{-1}$ be the i, j-th entry of the matrices $\boldsymbol{A}(\pi)$ and its inverse $\boldsymbol{A}^{-1}(\pi)$. This gives the following set of equations:

$$\sum_{j=1}^{d}A_{i,j}(\pi)A_{j,k}^{-1}(\pi) = \mathbb{1}\,(i = k), \qquad \text{for } i \in [d]. \tag{9}$$

Note that $\mathbb{1}\,(i = k)$ is a constant w.r.t. $\pi$. Hence,

$$\sum_{j=1}^{d}\left(\frac{dA_{i,j}(\pi)}{d\pi}A_{j,k}^{-1}(\pi) + \frac{dA_{j,k}^{-1}(\pi)}{d\pi}A_{i,j}(\pi)\right) = 0, \qquad \text{for } i \in [d]. \tag{10}$$

This gives

$$\frac{d\boldsymbol{A}^{-1}(\pi)}{d\pi} = -\boldsymbol{A}^{-1}\frac{d\boldsymbol{A}(\pi)}{d\pi}\boldsymbol{A}^{-1}. \tag{11}$$

We conclude the proof. $\qquad\square$

**Lemma 2.** *The $\boldsymbol{x}_t$-optimal design objective $v_t : \triangle_{\mathcal{X},\sigma} \to \mathbb{R}$ is differentiable with gradient at point $\boldsymbol{\pi} \in \triangle_{\mathcal{X},\sigma}$ given by*

$$\nabla_{\boldsymbol{\pi}}v_t(\boldsymbol{\pi}) = \left(-\left(\boldsymbol{x}_t^\mathsf{T}\left(\boldsymbol{V}(\boldsymbol{\pi})\right)^{-1}\boldsymbol{x}\right)^2\right)_{\boldsymbol{x}\in\mathcal{X}}. \tag{12}$$

*Proof.* Consider the partial derivative w.r.t. $\boldsymbol{x} \in \mathcal{X}$

$$\frac{\partial v_t(\boldsymbol{\pi})}{\partial \pi_{\boldsymbol{x}}} = \boldsymbol{x}_t^\mathsf{T}\frac{\partial\left(\boldsymbol{V}^{-1}(\boldsymbol{\pi})\right)}{\partial\pi_{\boldsymbol{x}}}\boldsymbol{x}_t = -\boldsymbol{x}_t^\mathsf{T}\left(\boldsymbol{V}^{-1}(\boldsymbol{\pi})\frac{\partial\boldsymbol{V}(\boldsymbol{\pi})}{\partial\pi_{\boldsymbol{x}}}\boldsymbol{V}^{-1}(\boldsymbol{\pi})\right)\boldsymbol{x}_t = -\boldsymbol{x}_t^\mathsf{T}\left(\boldsymbol{V}^{-1}(\boldsymbol{\pi})\boldsymbol{x}\boldsymbol{x}^\mathsf{T}\boldsymbol{V}^{-1}(\boldsymbol{\pi})\right)\boldsymbol{x}_t \tag{13}$$

$$= -\left(\boldsymbol{x}_t^\mathsf{T}\boldsymbol{V}^{-1}(\boldsymbol{\pi})\boldsymbol{x}\right)\left(\boldsymbol{x}_t^\mathsf{T}\boldsymbol{V}^{-1}(\boldsymbol{\pi})\boldsymbol{x}\right) = -\left(\boldsymbol{x}_t^\mathsf{T}\boldsymbol{V}^{-1}(\boldsymbol{\pi})\boldsymbol{x}\right)^2. \tag{14}$$

The first equality is obtained using Lemma 1 in the Appendix. This concludes the proof. $\qquad\square$

**Theorem 2.** *Suppose that Assumptions 1 and 3 hold. The $\boldsymbol{x}_t$-optimal design objective $v_t : \triangle_{\mathcal{X},\sigma} \to \mathbb{R}$ is convex, and $L_\sigma$-Lipschitz continuous w.r.t. the norm $\|\cdot\|_1$ for $L_\sigma \triangleq 4\frac{D_{\mathcal{X}}^2 D_{\mathcal{Z}}^2}{\sigma^2\omega^2}$.*

*Proof.* We first prove the convexity of $v_{t,\sigma}$. Let $\boldsymbol{X}, \boldsymbol{Y}$ be positive definite matrices. The inverse of a p.d. matrix is a convex operation. The following holds

$$(\lambda\boldsymbol{X} + (1-\lambda)\boldsymbol{Y})^{-1} \preceq \lambda\boldsymbol{X}^{-1} + (1-\lambda)\boldsymbol{Y}^{-1} \qquad \text{for } \lambda \in [0,1], \tag{15}$$

where $\boldsymbol{X} \preceq \boldsymbol{Y}$ denotes that $\boldsymbol{X} - \boldsymbol{Y}$ is positive semi definite. We have by considering $\lambda \in [0,1]$

$$v_t(\lambda\boldsymbol{\pi} + (1-\lambda)\boldsymbol{\pi}')) = \boldsymbol{x}_t^\mathsf{T}\left(\boldsymbol{V}(\lambda\boldsymbol{\pi} + (1-\lambda)\boldsymbol{\pi}'))\right)^{-1}\boldsymbol{x}_t = \boldsymbol{x}_t^\mathsf{T}\left(\boldsymbol{V}(\lambda\boldsymbol{\pi}) + \boldsymbol{V}((1-\lambda)\boldsymbol{\pi}')\right)^{-1}\boldsymbol{x}_t \tag{16}$$

$$\leq \lambda\boldsymbol{x}_t^\mathsf{T}\left(\boldsymbol{V}(\boldsymbol{\pi})\right)^{-1}\boldsymbol{x}_t + (1-\lambda)\boldsymbol{x}_t^\mathsf{T}\left(\boldsymbol{V}(\boldsymbol{\pi}')\right)^{-1}\boldsymbol{x}_t = \lambda v_t(\boldsymbol{\pi}) + (1-\lambda)v_t(\boldsymbol{\pi}'). \tag{17}$$

The equality follows from the definitions of $v_t$ and $\boldsymbol{V}(\boldsymbol{\pi})$. The inequality holds from the property of the inverse of positive definite matrices. This concludes the convexity proof.

Secondly, we prove the Lipschitzness of the function $v_t$. Let $\boldsymbol{\pi} \in (\tau+1)\triangle_{\mathcal{X},\sigma}$. We have the following from the gradient expression

$$\|\nabla_{\boldsymbol{\pi}} v_t(\boldsymbol{\pi})\|_\infty = \max_{\boldsymbol{x}\in\mathcal{X}} \left\{ \left(\boldsymbol{x}_t^\mathsf{T} \boldsymbol{V}^{-1}(\boldsymbol{\pi})\boldsymbol{x}\right)^2 \right\} \leq D_{\mathcal{Z}}^2 \max_{\boldsymbol{x}\in\mathcal{X}} \left\{ \left\|\boldsymbol{V}^{-1}(\boldsymbol{\pi})\boldsymbol{x}\right\|_2^2 \right\} \leq D_{\mathcal{Z}}^2 D_{\mathcal{X}}^2 \left\|\boldsymbol{V}^{-1}(\boldsymbol{\pi})\right\|_2^2$$

$$= D_{\mathcal{Z}}^2 D_{\mathcal{X}}^2 \left(\lambda_{\max}(\boldsymbol{V}^{-1}(\boldsymbol{\pi}))\right)^2 = D_{\mathcal{Z}}^2 D_{\mathcal{X}}^2 \left(\lambda_{\min}^{-1}(\boldsymbol{V}(\boldsymbol{\pi}))\right)^2 \leq \frac{4 D_{\mathcal{X}}^2 D_{\mathcal{Z}}^2}{\sigma^2 \omega^2} = \frac{4 D_{\mathcal{X}}^2 D_{\mathcal{Z}}^2}{\sigma^2 \omega^2}.$$

Eq. (1) follows from the gradient expression in Eq. (12). Eq. (2) is obtained using the Cauchy-Schwarz inequality. Eq. (3) and Eq. (4) follow from the definition of the spectral norm of a matrix. Eq. (5) uses the fact that the maximum eigenvalue of an invertible matrix is equal to the reciprocal of the minimum eigenvalue of its inverse. Finally, Eq. (6) follows from $\boldsymbol{V}(\boldsymbol{\pi}) \succeq \frac{\sigma}{\sigma+1}\omega \boldsymbol{I} \succeq \frac{\sigma\omega}{2}\boldsymbol{I}$, i.e., $\lambda_{\min}(\boldsymbol{V}(\boldsymbol{\pi})) \geq \lambda_{\min}\left(\frac{\sigma\omega}{2}\boldsymbol{I}\right) = \frac{\sigma\omega}{2}$. This part concludes the proof. $\qquad\square$

**Lemma 3.** *Let $\lambda_1, \lambda_2, \ldots, \lambda_T$ be a non-negative sequence of real numbers bounded by some constant, and let $\tau$ be a positive integer bounded by a constant. If the sum $\sum_{t=1}^T \lambda_t$ is $O(T^\alpha)$ for some $\alpha \in [0,1)$, then the sum $\sum_{t=\tau+1}^T \left(\sum_{s=t-\tau}^t \lambda_s\right)^2$ is $\mathcal{O}\left(T^\alpha\right)$.*

*Proof.* Consider the upper bound $\left(\sum_{s=t-\tau}^t \lambda_t\right)^2 \leq (\tau+1)\sum_{s=t-\tau}^t \lambda_t^2$. Apply Cauchy-Schwartz inequality to obtain

$$\left(\sum_{s=t-\tau}^t 1 \cdot \lambda_t\right)^2 \leq (\tau+1)\sum_{s=t-\tau}^t \lambda_t^2 \leq \left(\sum_{s=t-\tau}^t 1\right) \cdot \left(\sum_{s=t-\tau}^t \lambda_t^2\right). \tag{18}$$

Thus, it holds

$$\sum_{t=\tau+1}^T \left(\sum_{s=t-\tau}^t \lambda_t\right)^2 \leq (\tau+1)\sum_{t=\tau+1}^T \sum_{s=t-\tau}^t \lambda_t^2 \leq (\tau+1)^2 \sum_{t=1}^T \lambda_t^2. \tag{19}$$

It is sufficient to show that $\sum_{t=1}^T \lambda_t^2 = \mathcal{O}\left(T^\alpha\right)$ is true for $\sum_{t=\tau+1}^T \left(\sum_{s=t-\tau}^t \lambda_t\right)^2 = \mathcal{O}\left(T^\alpha\right)$ to be true.

Consider the following set $\mathcal{I} = \{t \in [T] : \lambda_t \geq 1\}$. We claim that $\sum_{t=1}^T \lambda_t = \mathcal{O}\left(T^\alpha\right)$ implies $|\mathcal{I}| = \mathcal{O}\left(T^\alpha\right)$. Assume that $|\mathcal{I}| = \Omega\left(T^{\alpha'}\right)$ and $\alpha' > \alpha$, but observe $\sum_{t=1}^T \lambda_t \geq \sum_{t\in\mathcal{I}} \lambda_t \geq \sum_{t\in\mathcal{I}} 1 = |\mathcal{I}|$ by the definition of $\mathcal{I}$. This means that $\sum_{t=1}^T \lambda_t = \Omega\left(T^{\alpha'}\right)$ which contradicts the assumption that $\sum_{t=1}^T \lambda_t = \mathcal{O}\left(T^\alpha\right)$. Therefore, we must have $|\mathcal{I}| = \mathcal{O}\left(T^\alpha\right)$.

Consider now the sum

$$\sum_{t=1}^T \lambda_t^2 = \sum_{t\in\mathcal{I}} \lambda_t^2 + \sum_{t\in[T]\setminus\mathcal{I}} \lambda_t^2 \leq |\mathcal{I}| \max\left\{\lambda_t^2 : t \in \mathcal{I}\right\} + \sum_{t\in[T]\setminus\mathcal{I}} \lambda_t = \mathcal{O}\left(T^\alpha\right). \tag{20}$$

Since $|\mathcal{I}| = \mathcal{O}\left(T^\alpha\right)$, the first sum is $\mathcal{O}\left(T^\alpha\right)$. The second sum is bounded by $\sum_{t\in[T]} \lambda_t$, which is $\mathcal{O}\left(T^\alpha\right)$ by the assumption that $\sum_{t=1}^T \lambda_t = \mathcal{O}\left(T^\alpha\right)$. Therefore, we have $\sum_{t=1}^T \lambda_t^2 = \mathcal{O}\left(T^\alpha\right)$. We conclude the proof. $\qquad\square$

**Lemma 4.** *Consider experimental designs $\boldsymbol{\pi}_1, \ldots, \boldsymbol{\pi}_T \in \triangle_{\mathcal{X}}$. For $t \in \{1, 2, ..., T-1\}$, define $\lambda_t = \|\boldsymbol{\pi}_t - \boldsymbol{\pi}_{t+1}\|_1$. Then, for any $\tau \in \{1, 2, ..., T\}$, the following inequality holds:*

$$\sum_{s=t-\tau}^t \|\boldsymbol{\pi}_s - \boldsymbol{\pi}_t\|_1 \leq (\tau+1)\sum_{s=t-\tau}^t \lambda_s. \tag{21}$$

*Proof.* Use the triangle inequality to obtain the following:

$$\sum_{s=t-\tau}^{t} \|\boldsymbol{\pi}_s - \boldsymbol{\pi}_t\|_1 \leq \sum_{s=t-\tau}^{t} \sum_{s'=s}^{t-1} \|\boldsymbol{\pi}_{s'+1} - \boldsymbol{\pi}_{s'}\|_1 = \sum_{s=t-\tau}^{t} \sum_{s'=s}^{t-1} \lambda_{s'} \tag{22}$$

$$\leq \sum_{s=t-\tau}^{t} \sum_{s'=t-\tau}^{t} \lambda_{s'} = (\tau+1) \sum_{t'=t-\tau}^{t} \lambda_{t'}. \tag{23}$$

The proof is concluded. □

**Lemma 5.** *Under Assumptions 1–3, a fixed design $\boldsymbol{\pi} \in \triangle_{\mathcal{X},\sigma}$, and a retention period $\tau \in \mathbb{N}$, the expected prediction error at time $t$ of the LSE model on experiment $\boldsymbol{x}_t \in \mathcal{Z}$ is*

$$\mathbb{E}\left[\left(y_t - \boldsymbol{x}_t \cdot \hat{\boldsymbol{\beta}}_t\right)^2\right] = \sigma^2 + \frac{\sigma^2 \boldsymbol{x}_t^{\mathsf{T}} \left(\boldsymbol{X}^{\mathsf{T}} \mathrm{diag}\left(\boldsymbol{\pi}\right) \boldsymbol{X}\right)^{-1} \boldsymbol{x}_t}{M(\tau+1)} + \left(\boldsymbol{x}_t \cdot \left(\boldsymbol{\beta}_t^{\star} - \frac{1}{\tau+1} \sum_{s=t-\tau}^{t} \boldsymbol{\beta}_s^{\star}\right)\right)^2.$$

*Proof.* We can rewrite Eq. (1) as follows. It is easy to see that $\boldsymbol{x}_t^{\mathsf{T}} \left(\boldsymbol{X}^{\mathsf{T}} \mathrm{diag}\left((\tau+1)\boldsymbol{\pi}\right) \boldsymbol{X}\right)^{-1} \boldsymbol{x}_t = \frac{\boldsymbol{x}_t^{\mathsf{T}} \boldsymbol{V}^{-1}(\boldsymbol{\pi})\boldsymbol{x}_t}{\tau+1}$. Also, it holds

$$\frac{1}{\tau+1} \left(\boldsymbol{X}^{\mathsf{T}} \mathrm{diag}\left(\boldsymbol{\pi}\right) \boldsymbol{X}\right)^{-1} \left(\sum_{\boldsymbol{x} \in \mathcal{X}} \boldsymbol{x}\boldsymbol{x}^{\mathsf{T}} \sum_{s=t-\tau}^{t} \pi_{\boldsymbol{x}} \boldsymbol{\beta}_s^{\star}\right) \tag{24}$$

$$= \left(\boldsymbol{X}^{\mathsf{T}} \mathrm{diag}\left(\boldsymbol{\pi}\right) \boldsymbol{X}\right)^{-1} \left(\sum_{\boldsymbol{x} \in \mathcal{X}} \boldsymbol{x}\boldsymbol{x}^{\mathsf{T}} \pi_{\boldsymbol{x}} \left(\frac{1}{\tau+1} \sum_{s=t-\tau}^{t} \boldsymbol{\beta}_s^{\star}\right)\right) \tag{25}$$

$$= \left(\boldsymbol{X}^{\mathsf{T}} \mathrm{diag}\left(\boldsymbol{\pi}\right) \boldsymbol{X}\right)^{-1} \left(\boldsymbol{X}^{\mathsf{T}} \mathrm{diag}\left(\boldsymbol{\pi}\right) \boldsymbol{X}\right) \left(\frac{1}{\tau+1} \sum_{s=t-\tau}^{t} \boldsymbol{\beta}_s^{\star}\right) = \boldsymbol{I} \left(\frac{1}{\tau+1} \sum_{s=t-\tau}^{t} \boldsymbol{\beta}_s^{\star}\right) \tag{26}$$

$$= \left(\frac{1}{\tau+1} \sum_{s=t-\tau}^{t} \boldsymbol{\beta}_s^{\star}\right). \tag{27}$$

This concludes the proof. □

**Lemma 6.** *Under Assumptions 1–3, let $\{\boldsymbol{\pi}_s\}_{s=t-\tau}^{t} \in \triangle_{\mathcal{X},\sigma}^{t}$ be a sequence of designs, and let $\lambda_t = \|\boldsymbol{\pi}_t - \boldsymbol{\pi}_{t+1}\|_1$. Then, the expected prediction error of the LSE model on experiment $\boldsymbol{x}_t \in \mathcal{Z}$ at time $t$ under an inference window size $\tau$ is bounded as follows:*

$$|f_t(\boldsymbol{\pi}_t, \ldots, \boldsymbol{\pi}_t) - f_t(\boldsymbol{\pi}_{t-\tau}, \ldots, \boldsymbol{\pi}_t)| \leq \epsilon_{\boldsymbol{\lambda},t,\tau}, \tag{28}$$

*where* $\epsilon_{\boldsymbol{\lambda},t,\tau} = 8 \left(\frac{|\mathcal{X}| D_{\mathcal{Z}} B_{\mathcal{X} \cup \mathcal{Z}} D_{\mathcal{X}}}{\omega \sigma} \sum_{t'=t-\tau}^{t} \lambda_{t'}\right)^2 + 16 \frac{|\mathcal{X}| D_{\mathcal{Z}} B_{\mathcal{X} \cup \mathcal{Z}}^2 D_{\mathcal{X}}}{\omega \sigma} \sum_{t'=t-\tau}^{t} \lambda_{t'} + \frac{4 D_{\mathcal{X}}^2 D_{\mathcal{Z}}^2}{\sigma^2 \omega^2 M} \sum_{t'=t-\tau}^{t} \lambda_{t'}.$

*Proof.* The proof is divided into two parts. In the first part we bound the variance term, and in the second part we bound the bias term.

**Part 1.** We bound the quantity $|v_t(\boldsymbol{\pi}_{t-\tau}, \ldots, \boldsymbol{\pi}_t) - v_t(\boldsymbol{\pi}_t, \ldots, \boldsymbol{\pi}_t)|$. Lemma 2 shows that $v_t$ is Lipschitz continuous over $\triangle_{\mathcal{X},\sigma}$ with parameter $L_\sigma = \frac{4D_{\mathcal{X}}^2 D_{\mathcal{Z}}^2}{\sigma^2 \omega^2 M(\tau+1)}$. Thus, we have the following:

$$|v_t(\boldsymbol{\pi}_{t-\tau}, \ldots, \boldsymbol{\pi}_t) - v_t(\boldsymbol{\pi}_t, \ldots, \boldsymbol{\pi}_t)| = |v_t(\boldsymbol{\pi}_{t-\tau:t}) - v_t((\tau+1)\boldsymbol{\pi}_t)| \tag{29}$$

$$\leq L_\sigma \|\boldsymbol{\pi}_{t-\tau:t} - (\tau+1)\boldsymbol{\pi}_t\|_1 \tag{30}$$

$$\leq L_\sigma \sum_{s=t-\tau}^t \|\boldsymbol{\pi}_s - \boldsymbol{\pi}_t\|_1 \tag{31}$$

$$\leq L_\sigma(\tau+1) \sum_{t'=t-\tau}^t \lambda_{t'} \qquad \text{(Lemma 4)} \tag{32}$$

$$= \frac{4D_{\mathcal{X}}^2 D_{\mathcal{Z}}^2}{\sigma^2 \omega^2 M} \left( \sum_{t'=t-\tau}^t \lambda_{t'} \right). \qquad \text{(Lemma 2)} \tag{33}$$

**Part 2.** We bound the quantity $|b_t(\boldsymbol{\pi}_{t-\tau}, \ldots, \boldsymbol{\pi}_t) - b_t(1/|\mathcal{X}|, \ldots, 1/|\mathcal{X}|)|$. Recall that

$$b_t(\boldsymbol{\pi}_{t-\tau}, \ldots, \boldsymbol{\pi}_t) = \left( \boldsymbol{x}_t \cdot \left( \boldsymbol{V}^{-1}(\boldsymbol{\pi}_{t-\tau:t}) \left( \sum_{\boldsymbol{x} \in \mathcal{X}} \boldsymbol{x}\boldsymbol{x}^\mathsf{T} \sum_{s=t-\tau}^t \pi_{s,\boldsymbol{x}}\boldsymbol{\beta}_s^\star \right) - \boldsymbol{\beta}_t^\star \right) \right)^2. \tag{34}$$

Observe the following:

$$\mathbb{E}\left[ \boldsymbol{x}_t \cdot \hat{\boldsymbol{\beta}}_t \right] = \boldsymbol{x}_t \cdot \left( \boldsymbol{V}^{-1}(\boldsymbol{\pi}_{t-\tau:t}) \left( M \sum_{\boldsymbol{x} \in \mathcal{X}} \boldsymbol{x}\boldsymbol{x}^\mathsf{T} \sum_{s=t-\tau}^t \pi_{s,\boldsymbol{x}}\boldsymbol{\beta}_s^\star \right) \right) \tag{35}$$

$$= \boldsymbol{x}_t \cdot \boldsymbol{V}^{-1}(\boldsymbol{\pi}_{t-\tau:t}) \left( M \sum_{\boldsymbol{x} \in \mathcal{X}} \boldsymbol{x}\boldsymbol{x}^\mathsf{T} \frac{\pi_{t-\tau:t,\boldsymbol{x}}}{\tau+1} \sum_{s=t-\tau}^t \boldsymbol{\beta}_s^\star \right) + \boldsymbol{x}_t \cdot (\boldsymbol{\Delta}_1 + \boldsymbol{\Delta}_2), \tag{36}$$

where

$$\boldsymbol{\Delta}_1 = \boldsymbol{V}^{-1}(\boldsymbol{\pi}_{t-\tau:t}) \left( M \sum_{\boldsymbol{x} \in \mathcal{X}} \boldsymbol{x}\boldsymbol{x}^\mathsf{T} \sum_{s=t-\tau}^t (\pi_{s,\boldsymbol{x}} - \pi_{t,\boldsymbol{x}})\boldsymbol{\beta}_s^\star \right), \tag{37}$$

$$\boldsymbol{\Delta}_2 = \boldsymbol{V}^{-1}(\boldsymbol{\pi}_{t-\tau:t}) \left( M \sum_{\boldsymbol{x} \in \mathcal{X}} \boldsymbol{x}\boldsymbol{x}^\mathsf{T} \frac{(\tau+1)\pi_{t,\boldsymbol{x}} - \pi_{t-\tau:\tau,\boldsymbol{x}}}{\tau+1} \sum_{s=t-\tau}^t \boldsymbol{\beta}_s^\star \right). \tag{38}$$

Moreover, note that the first term in Eq. (36) satisfies

$$\boldsymbol{x}_t \cdot \left( \boldsymbol{V}^{-1}(\boldsymbol{\pi}_{t-\tau:t}) \left( M \sum_{\boldsymbol{x} \in \mathcal{X}} \boldsymbol{x}\boldsymbol{x}^\mathsf{T} \frac{\pi_{t-\tau:t,\boldsymbol{x}}}{\tau+1} \sum_{s=t-\tau}^t \boldsymbol{\beta}_s^\star \right) \right) \tag{39}$$

$$= \boldsymbol{x}_t \cdot \left( \boldsymbol{V}^{-1}(\boldsymbol{\pi}_{t-\tau:t})\boldsymbol{V}(\boldsymbol{\pi}_{t-\tau:t}) \left( \frac{1}{\tau+1} \sum_{s=t-\tau}^t \boldsymbol{\beta}_s^\star \right) \right) = \boldsymbol{x}_t \cdot \left( \frac{1}{\tau+1} \sum_{s=t-\tau}^t \boldsymbol{\beta}_s^\star \right). \tag{40}$$

We establish upper bounds on the remaining terms:

$$|\boldsymbol{x}_t \cdot \boldsymbol{\Delta}_1| \leq \sum_{\boldsymbol{x} \in \mathcal{X}} \sum_{s=t-\tau}^t \left| \boldsymbol{V}^{-1}(\boldsymbol{\pi}_{t-\tau:t}) \left( M\boldsymbol{x}\boldsymbol{x}^\mathsf{T} ((\tau+1)\pi_{t,\boldsymbol{x}} - \pi_{t-\tau:\tau,\boldsymbol{x}})\boldsymbol{\beta}_s^\star \right) \cdot \boldsymbol{x}_t \right| \tag{41}$$

$$\leq \sum_{\boldsymbol{x} \in \mathcal{X}} \sum_{s=t-\tau}^t D_{\mathcal{Z}} \left\| \boldsymbol{V}^{-1}(\boldsymbol{\pi}_{t-\tau:t}) \left( M\boldsymbol{x}\boldsymbol{x}^\mathsf{T} ((\tau+1)\pi_{t,\boldsymbol{x}} - \pi_{t-\tau:\tau,\boldsymbol{x}})\boldsymbol{\beta}_s^\star \right) \right\|_2 \tag{42}$$

$$\leq 2 \sum_{\boldsymbol{x} \in \mathcal{X}} \frac{D_{\mathcal{Z}}}{\omega\sigma(\tau+1)} B_{\mathcal{X} \cup \mathcal{Z}} D_{\mathcal{X}} |(\tau+1)\pi_{t,\boldsymbol{x}} - \pi_{t-\tau:\tau,\boldsymbol{x}}| \tag{43}$$

$$\leq \frac{2|\mathcal{X}| D_{\mathcal{Z}} B_{\mathcal{X} \cup \mathcal{Z}} D_{\mathcal{X}}}{\omega\sigma} \sum_{t'=t-\tau}^t \lambda_{t'}. \tag{44}$$

Following the same steps, we obtain

$$|\boldsymbol{x}_t \cdot \boldsymbol{\Delta}_2| \leq \sum_{\boldsymbol{x} \in \mathcal{X}} \sum_{s=t-\tau}^{t} \frac{2 D_{\mathcal{Z}} B_{\mathcal{X} \cup \mathcal{Z}} D_{\mathcal{X}}}{\omega \sigma (\tau+1)} |\pi_{s,\boldsymbol{x}} - \pi_{t,\boldsymbol{x}}| \leq \frac{2 |\mathcal{X}| D_{\mathcal{Z}} B_{\mathcal{X} \cup \mathcal{Z}} D_{\mathcal{X}}}{\omega \sigma} \sum_{t'=t-\tau}^{t} \lambda_{t'}. \quad (45)$$

Use the simple fact that $(x+y)^2 = x^2 + y^2 + 2xy$, and $(x+y)^2 \leq 2x^2 + 2y^2$ to obtain

$$|b_t(\boldsymbol{\pi}_{t-\tau}, \ldots, \boldsymbol{\pi}_t) - b_t(\boldsymbol{1}/|\mathcal{X}|, \ldots, \boldsymbol{1}/|\mathcal{X}|)| \quad (46)$$

$$= \left| \left( \boldsymbol{x}_t \cdot \left( \mathbb{E}\left[\hat{\boldsymbol{\beta}}_t\right] - \boldsymbol{\beta}_t^\star \right) \right)^2 - \left( \boldsymbol{x}_t \cdot \left( \frac{1}{\tau+1} \sum_{s=t-\tau}^{t} \boldsymbol{\beta}_s^\star - \boldsymbol{\beta}_t^\star \right) \right)^2 \right| \quad (47)$$

$$= \left| \left( \boldsymbol{x}_t \cdot \left( \left( \frac{1}{\tau+1} \sum_{s=t-\tau}^{t} \boldsymbol{\beta}_s^\star + \boldsymbol{\Delta}_1 + \boldsymbol{\Delta}_2 \right) - \boldsymbol{\beta}_t^\star \right) \right)^2 - \left( \boldsymbol{x}_t \cdot \left( \frac{1}{\tau+1} \sum_{s=t-\tau}^{t} \boldsymbol{\beta}_s^\star - \boldsymbol{\beta}_t^\star \right) \right)^2 \right| \quad (48)$$

$$= \left| \left( (\boldsymbol{x}_t \cdot (\boldsymbol{\Delta}_1 + \boldsymbol{\Delta}_2))^2 + 2 (\boldsymbol{x}_t \cdot (\boldsymbol{\Delta}_1 + \boldsymbol{\Delta}_2)) \left( \boldsymbol{x}_t \cdot \left( \frac{1}{\tau+1} \sum_{s=t-\tau}^{t} \boldsymbol{\beta}_s^\star \right) - \boldsymbol{\beta}_t^\star \right) \right) \right| \quad (49)$$

$$\leq 8 \left( \frac{|\mathcal{X}| D_{\mathcal{Z}} B_{\mathcal{X} \cup \mathcal{Z}} D_{\mathcal{X}}}{\omega \sigma} \sum_{t'=t-\tau}^{t} \lambda_{t'} \right)^2 + 16 \frac{|\mathcal{X}| D_{\mathcal{Z}} B_{\mathcal{X} \cup \mathcal{Z}}^2 D_{\mathcal{X}}}{\omega \sigma} \sum_{t'=t-\tau}^{t} \lambda_{t'}. \quad (50)$$

Finally, combine Eqs. (33) and (50)

$$|f_t(\boldsymbol{\pi}_t, \ldots, \boldsymbol{\pi}_t) - f_t(\boldsymbol{\pi}_{t-\tau}, \ldots, \boldsymbol{\pi}_t)| \quad (51)$$

$$\leq 8 \left( \frac{|\mathcal{X}| D_{\mathcal{Z}} B_{\mathcal{X} \cup \mathcal{Z}} D_{\mathcal{X}}}{\omega \sigma} \sum_{t'=t-\tau}^{t} \lambda_{t'} \right)^2 + 16 \frac{|\mathcal{X}| D_{\mathcal{Z}} B_{\mathcal{X} \cup \mathcal{Z}}^2 D_{\mathcal{X}}}{\omega \sigma} \sum_{t'=t-\tau}^{t} \lambda_{t'} + \frac{4 D_{\mathcal{X}}^2 D_{\mathcal{Z}}^2}{\sigma^2 \omega^2 M} \left( \sum_{t'=t-\tau}^{t} \lambda_{t'} \right). \quad (52)$$

We conclude the proof. □

**Proposition 3.** *Under Assumptions 1–3, let $\{\boldsymbol{\pi}_s\}_{s=t-\tau}^{t} \in \triangle_{\mathcal{X},\sigma}$ be a sequence of designs, and let $P_T = \sum_{t=1}^{T} \|\boldsymbol{\pi}_t - \boldsymbol{\pi}_{t+1}\|_1$. Then, the EPE of the LSE on sequence of experiments $\{\boldsymbol{x}_t\}_{t=1}^{T} \in \mathcal{Z}$ at time t under an inference window size $\tau$ is bounded as follows:*

$$\sum_{t=\tau+1}^{T} |f_t(\boldsymbol{\pi}_t, \ldots, \boldsymbol{\pi}_t) - f_t(\boldsymbol{\pi}_{t-\tau}, \ldots, \boldsymbol{\pi}_t)| = \mathcal{O}\left( P_T \right). \quad (53)$$

*Proof.* From Lemma 6 we have

$$\sum_{t=\tau+1}^{T} |f_t(\boldsymbol{\pi}_t, \ldots, \boldsymbol{\pi}_t) - f_t(\boldsymbol{\pi}_{t-\tau}, \ldots, \boldsymbol{\pi}_t)| = \sum_{t=\tau+1}^{T} \epsilon_{\boldsymbol{\lambda},t,\tau} = \mathcal{O}\left( \sum_{t=\tau+1}^{T} \lambda_t + \sum_{t=\tau+1}^{T} \left( \sum_{st-\tau}^{t} \lambda_s \right)^2 \right) \quad (54)$$

$$= \mathcal{O}\left( P_T \right) + \mathcal{O}\left( \sum_{t=\tau+1}^{T} \left( \sum_{st-\tau}^{t} \lambda_s \right)^2 \right). \quad (55)$$

Lemma 3 gives $\mathcal{O}\left( \sum_{t=\tau+1}^{T} \left( \sum_{st-\tau}^{t} \lambda_s \right)^2 \right) = \mathcal{O}\left( P_T \right)$. This concludes the proof. □

## C  A Unified Analysis via Mirror Descent Schemes

**Mirror Descent Parametrization.** We provide the set of assumptions relevant to correct parametrization of the online mirror descent family of gradient-based policies.

**Assumption 4.** *The map $\Phi : \mathcal{S}_\Phi \to \mathbb{R}$ satisfies the following properties:*

- *The domain $\mathcal{S}_\Phi$ of $\Phi$ is a convex and open set such that the decision set $\mathcal{S}$ is included in its closure, i.e., $\mathcal{S} \subseteq \mathrm{closure}(\mathcal{D})$, and their intersection is nonempty $\mathcal{S} \cap \mathcal{S}_\Phi \neq \emptyset$.*

- *The map $\Phi$ is $\rho$ strongly-convex over $\mathcal{S}_\Phi$ w.r.t. a norm $\|\cdot\|$ and differentiable over $\mathcal{S}_\Phi$.*

- *The map $\nabla\Phi(\boldsymbol{\pi}) : \mathcal{S}_\Phi \to \mathbb{R}^n$ is surjective.*

- *The gradient of $\Phi$ diverges on the boundary of $\mathcal{S}_\Phi$, i.e., $\lim_{\boldsymbol{\pi} \to \partial\mathcal{S}_\Phi} \|\nabla\Phi(\boldsymbol{\pi})\| = +\infty$, where $\partial\mathcal{S}_\Phi = \mathrm{closure}(\mathcal{S}_\Phi) \setminus \mathcal{S}_\Phi$.*

A map $\Phi : \mathcal{S}_\Phi \to \mathbb{R}$ is said to be a mirror map if it satisfies Assumption 4.

**Assumption 5.** *Consider a decision set $\mathcal{S}$ and a mirror map $\Phi$. The dual norm $\|\cdot\|_\star$ of the gradient of the mirror map $\Phi$ is bounded by $L_\Phi \in \mathbb{R}_{\geq 0}$, i.e., $\|\nabla\Phi(\boldsymbol{\pi})\|_\star \leq L_\Phi$ for every $\boldsymbol{\pi} \in \mathcal{S}$.*

**Assumption 6.** *Consider a decision set $\mathcal{S}$, a map $\Phi$, and $\boldsymbol{\pi}_1 = \mathrm{argmin}_{\mathcal{S}}\Phi(\boldsymbol{\pi})$. The Bregman divergence $D_\Phi$ is bounded over $\mathcal{S}$, i.e., there exits $D_{\Phi,\max}$ s.t. $D_\Phi(\boldsymbol{\pi}, \boldsymbol{\pi}_1) \leq D_{\Phi,\max}^2 < \infty$ for any $\boldsymbol{\pi} \in \mathcal{S}$.*

**Definitions.** We present formal definitions that are crucial for our subsequent analysis of regret bounds associated with mirror descent algorithms.

**Definition 3.** *The Bregman projection [29] associated to a map $\Phi$ onto a convex set $\mathcal{S}$ is denoted by $\Pi_{\mathcal{S}}^\Phi : \mathbb{R}^n \to \mathcal{S}$, is defined as*

$$\Pi_{\mathcal{S}}^\Phi(\boldsymbol{\pi}') = \arg\min_{\boldsymbol{\pi} \in \mathcal{S}} D_\Phi(\boldsymbol{\pi}, \boldsymbol{\pi}'), \quad \text{where} \quad D_\Phi(\boldsymbol{\pi}, \boldsymbol{\pi}') = \Phi(\boldsymbol{\pi}) - \Phi(\boldsymbol{\pi}') - \nabla\Phi(\boldsymbol{\pi}') \cdot (\boldsymbol{\pi} - \boldsymbol{\pi}'). \quad (56)$$

**Definition 4.** *Let $\Phi : \mathcal{S}_\Phi \to \mathbb{R}$ be a mirror map satisfying Assumption 4, and $\eta \in \mathbb{R}_{>0}$ be the learning rate. At timeslot $t \in [T]$, Online Mirror Descent upon receiving cost function $f_t$ it updates the decision $\boldsymbol{\pi}_t$ according to the mapping*

$$\boldsymbol{\pi}_{t+1} = \Pi_{\mathcal{S} \cap \mathcal{S}_\Phi}\left((\nabla\Phi)^{-1}\left(\nabla\Phi(\boldsymbol{\pi}_t) - \eta\nabla_{\boldsymbol{\pi}}f_t(\boldsymbol{\pi}_t)\right)\right). \quad (57)$$

### C.1 Technical Lemmas

**Lemma 7.** *(First Order Optimality Condition) Let $f : \mathcal{S} \to \mathbb{R}$ be convex and $\mathcal{S}$ a closed convex set on which $f$ is differentiable. Then*

$$\boldsymbol{\pi}^\star \in \mathrm{argmin}_{\boldsymbol{\pi} \in \mathcal{S}} f(\boldsymbol{\pi}) \iff \nabla f(\boldsymbol{\pi}^\star) \cdot (\boldsymbol{\pi}^\star - \boldsymbol{\pi}'), \forall \boldsymbol{\pi}' \in \mathcal{S}. \quad (58)$$

**Lemma 8.** *Assume that $\Phi$ is $\rho$-strongly convex w.r.t $\|\cdot\|$. Let $\|\cdot\|_\star$ be the dual norm of $\|\cdot\|$ and $\eta \in \mathbb{R}_{>0}$, and $\boldsymbol{g}_t \in \mathbb{R}^d$ be the gradient at time $t$. The upper bound $\Lambda(\boldsymbol{\pi}_t, \boldsymbol{g}_t) \geq \frac{D_\Phi(\boldsymbol{\pi}_t, \boldsymbol{z}_{t+1})}{\eta^2}$ can be taken as*

$$\Lambda(\boldsymbol{\pi}_t, \boldsymbol{g}_t) = \frac{\|\boldsymbol{g}_t\|_\star^2}{2\rho}. \quad (59)$$

*Proof.* Expand $D_\Phi(\boldsymbol{\pi}_t, \boldsymbol{z}_{t+1})$ to obtain

$$D_\Phi(\boldsymbol{\pi}_t, \boldsymbol{z}_{t+1}) = \Phi(\boldsymbol{\pi}_t) - \Phi(\boldsymbol{z}_{t+1}) - \nabla\Phi(\boldsymbol{z}_{t+1}) \cdot (\boldsymbol{\pi}_t - \boldsymbol{z}_{t+1}) \quad (60)$$

$$= \Phi(\boldsymbol{\pi}_t) - \Phi(\boldsymbol{z}_{t+1}) + \nabla\Phi(\boldsymbol{\pi}_t) \cdot (\boldsymbol{z}_{t+1} - \boldsymbol{\pi}_t) + \nabla\Phi(\boldsymbol{\pi}_t) - \nabla\Phi(\boldsymbol{z}_{t+1}) \cdot (\boldsymbol{\pi}_t - \boldsymbol{z}_{t+1}) \quad (61)$$

$$\leq -\frac{\rho}{2}\|\boldsymbol{\pi}_t - \boldsymbol{z}_{t+1}\|^2 + \eta\boldsymbol{g}_t \cdot (\boldsymbol{\pi}_t - \boldsymbol{z}_{t+1}). \quad (62)$$

By the strong convexity of $\Phi$ and the gradient step. Use Cauchy-Schwarz inequality to obtain

$$D_\Phi(\boldsymbol{\pi}_t, \boldsymbol{z}_{t+1}) \leq \eta\|\boldsymbol{g}_t\|_\star\|\boldsymbol{\pi}_t - \boldsymbol{z}_{t+1}\| - \frac{\rho}{2}\|\boldsymbol{\pi}_t - \boldsymbol{z}_{t+1}\|^2 \leq \frac{\eta^2\|\boldsymbol{g}_t\|_\star^2}{2\rho}, \quad (63)$$

since $\max_z(az - bz^2) = a^2/4b$. Combine (63) and (62) to conclude the proof. $\qquad\square$

**Lemma 9.** *When $\Phi$ is the negative entropy, $\boldsymbol{\pi} \in \mathbb{R}^n_{>0}$, and $\boldsymbol{g}_t \in \mathbb{R}^n_{\geq 0}$. The upper bound $\Lambda(\boldsymbol{\pi}_t, \boldsymbol{g}_t) \geq \frac{D_\Phi(\boldsymbol{\pi}_t, \boldsymbol{z}_{t+1})}{\eta^2}$ can be taken as*

$$\Lambda(\boldsymbol{\pi}_t, \boldsymbol{g}_t) = \sum_{i=1}^n \frac{x_i (g_{t,i})^2}{2}. \tag{64}$$

*Proof.* Expand $D_\Phi(\boldsymbol{\pi}_t, \boldsymbol{z}_{t+1})$ to obtain

$$D_\Phi(\boldsymbol{\pi}_t, \boldsymbol{z}_{t+1}) = \sum_{i=1}^n x_{t,i} \left(\exp(-\eta g_{t,i}) + \eta g_{t,i} - 1\right). \tag{65}$$

Since $\exp(x) - x - 1 \leq x^2/2$ for $x \leq 0$, it holds $D_\Phi(\boldsymbol{\pi}_t, \boldsymbol{z}_{t+1}) \leq \eta^2 \sum_{i=1}^n x_{t,i}(g_{t,i})^2/2 = \eta^2 \Lambda(\boldsymbol{\pi}_t, \boldsymbol{g}_t)$. We conclude the proof. $\square$

**Lemma 10.** *Fix $\boldsymbol{z} \in \mathcal{D}$, and let $\boldsymbol{\pi} = \Pi_\mathcal{X}^\Phi(\boldsymbol{z})$. Then*

$$D_\Phi(\boldsymbol{\pi}', \boldsymbol{z}) \geq D_\Phi(\boldsymbol{\pi}', \boldsymbol{\pi}) \qquad \forall \boldsymbol{\pi}' \in \mathcal{X}. \tag{66}$$

*Proof.* The generalized Pythagorean equality given by

$$D_\Phi(\boldsymbol{\pi}, \boldsymbol{\pi}') + D_\Phi(\boldsymbol{\pi}'', \boldsymbol{\pi}) - D_\Phi(\boldsymbol{\pi}'', \boldsymbol{\pi}') = (\nabla\Phi(\boldsymbol{\pi}) - \nabla\Phi(\boldsymbol{\pi}')) \cdot (\boldsymbol{\pi} - \boldsymbol{\pi}'') \tag{67}$$

from the definition of the Bregman divergence, and the first order optimality condition [8], the following holds

$$D_\Phi(\boldsymbol{\pi}, \boldsymbol{z}) + D_\Phi(\boldsymbol{\pi}', \boldsymbol{\pi}) - D_\Phi(\boldsymbol{\pi}', \boldsymbol{z}) = (\nabla\Phi(\boldsymbol{\pi}) - \nabla\Phi(\boldsymbol{z})) \cdot (\boldsymbol{\pi} - \boldsymbol{\pi}') \leq 0. \tag{68}$$

The proof is concludes by noting that $D_\Phi(\boldsymbol{\pi}', \boldsymbol{z}) \geq 0$ for any $\boldsymbol{\pi}' \in \mathcal{D}$. $\square$

**Lemma 11.** *At time $t$ under a gradient $\boldsymbol{g}_t$, OMD update rule with state $\boldsymbol{\pi}_t$ satisfies the following*

$$\boldsymbol{g}_t \cdot (\boldsymbol{\pi}_t - \boldsymbol{\pi}) \leq \frac{1}{\eta} \left(\eta^2 \Lambda(\boldsymbol{g}_t, \boldsymbol{\pi}_t) + D_\Phi(\boldsymbol{\pi}, \boldsymbol{\pi}_t) - D_\Phi(\boldsymbol{\pi}, \boldsymbol{\pi}_{t+1})\right). \tag{69}$$

*Proof.* By the gradient step, $\boldsymbol{g}_t = (\boldsymbol{\pi}_t - \boldsymbol{z}_{t+1})/\eta$, so

$$\boldsymbol{g}_t \cdot (\boldsymbol{\pi}_t - \boldsymbol{\pi}) = (\nabla\Phi(\boldsymbol{\pi}_t) - \nabla\Phi(\boldsymbol{z}_{t+1})) (\boldsymbol{\pi}_t - \boldsymbol{\pi}) \tag{70}$$

$$= \frac{1}{\eta} \left(D_\Phi(\boldsymbol{\pi}_t, \boldsymbol{z}_{t+1}) + D_\Phi(\boldsymbol{\pi}, \boldsymbol{\pi}_t) - D_\Phi(\boldsymbol{\pi}, \boldsymbol{z}_{t+1})\right) \tag{71}$$

$$\leq \frac{1}{\eta} \left(D_\Phi(\boldsymbol{\pi}_t, \boldsymbol{z}_{t+1}) + D_\Phi(\boldsymbol{\pi}, \boldsymbol{\pi}_t) - D_\Phi(\boldsymbol{\pi}, \boldsymbol{\pi}_{t+1})\right) \tag{72}$$

$$\tag{73}$$

The first equality is obtained using Eq (67). The inequality is obtained using Lemma 10. This concludes the proof. $\square$

## C.2 Regret Guarantee

**Theorem 3.** *Consider $\mathcal{S} \subseteq \mathbb{R}^d$ as the decision set, and a comparator sequence $\{\boldsymbol{\pi}_t^\star\}_{t=1}^T \in \mathcal{S}^T$ with path-length $P_T = \sum_{t=1}^T \|\boldsymbol{\pi}_t^\star - \boldsymbol{\pi}_{t+1}^\star\|_1$. Under Assumptions (4), and (5), Online Mirror Descent (57) configured with mirror map $\Phi : \mathcal{S}_\Phi \to \mathbb{R}$ and learning rate $\eta \in \mathbb{R}_{\geq 0}$ has the following regret guarantee against L-Lipschitz (w.r.t. $\|\cdot\|$) differentiable convex cost functions $f_1, \ldots, f_T$:*

$$\sum_{t=1}^T f_t(\boldsymbol{\pi}_t) - \sum_{t=1}^T f_t(\boldsymbol{\pi}_t^\star) \leq \frac{1}{\eta} \left(D_{\Phi,\max}^2 + 2L_\Phi P_T\right) + \eta \left(\frac{L^2 T}{2\rho}\right). \tag{74}$$

*The policy-induced decisions exhibit a path length of $\sum_{t=1}^T \|\boldsymbol{\pi}_{t+1} - \boldsymbol{\pi}_t\| = \mathcal{O}(\eta T)$.*

*Proof.* The cost functions $f_t$ are convex and differentiable. So, it holds $\sum_{t=1}^{T} f_t(\boldsymbol{\pi}_t) - \sum_{t=1}^{T} f_t(\boldsymbol{\pi}_t^\star) \leq \nabla_{\boldsymbol{\pi}} f_t(\boldsymbol{\pi}_t) \cdot (\boldsymbol{\pi}_t - \boldsymbol{\pi}_t^\star)$. Lemma 11 gives:

$$\sum_{t=1}^{T} f_t(\boldsymbol{\pi}_t) - \sum_{t=1}^{T} f_t(\boldsymbol{\pi}_t^\star) \leq \sum_{t=1}^{T} \frac{1}{\eta} \left( D_\Phi(\boldsymbol{\pi}, \boldsymbol{\pi}_t) - D_\Phi(\boldsymbol{\pi}, \boldsymbol{\pi}_{t+1}) \right) + \eta \sum_{t=1}^{T} \Lambda \left( \nabla_{\boldsymbol{\pi}} f_t(\boldsymbol{\pi}_t) \right). \tag{75}$$

Bounding the terms $D_\Phi(\boldsymbol{\pi}, \boldsymbol{\pi}_t) - D_\Phi(\boldsymbol{\pi}, \boldsymbol{\pi}_{t+1})$:

$$\sum_{t \in [T]} \left( D_\Phi(\boldsymbol{\pi}_t^\star, \boldsymbol{\pi}_{t-1}) - D_\Phi(\boldsymbol{\pi}_t^\star, \boldsymbol{\pi}_t) \right) \leq D_\Phi(\boldsymbol{\pi}_1^\star, \boldsymbol{\pi}_0) + \sum_{t \in [T]} \left( D_\Phi(\boldsymbol{\pi}_{t+1}^\star, \boldsymbol{\pi}_t) - D_\Phi(\boldsymbol{\pi}_t^\star, \boldsymbol{\pi}_t) \right) \tag{76}$$

$$\leq D_{\Phi,\max}^2 + \sum_{t \in [T]} \underbrace{\left( \nabla \Phi(\boldsymbol{\pi}_{t+1}^\star) - \nabla \Phi(\boldsymbol{\pi}_t) \right) \cdot \left( \boldsymbol{\pi}_{t+1}^\star - \boldsymbol{\pi}_t^\star \right)}_{\text{Cauchy-Schwarz's Ineq.}} - \underbrace{D_\Phi(\boldsymbol{\pi}_t^\star, \boldsymbol{\pi}_{t+1}^\star)}_{\geq 0} \tag{77}$$

$$\leq D_{\Phi,\max}^2 + \sum_{t \in [T]} \underbrace{\left\| \nabla \Phi(\boldsymbol{\pi}_{t+1}^\star) - \nabla \Phi(\boldsymbol{\pi}_t) \right\|_\star}_{\leq L_\Phi} \left\| \boldsymbol{\pi}_{t+1}^\star - \boldsymbol{\pi}_t^\star \right\| \tag{78}$$

$$\leq D_{\Phi,\max}^2 + 2L_\Phi \sum_{t \in [T]} \left\| \boldsymbol{\pi}_{t+1}^\star - \boldsymbol{\pi}_t^\star \right\|. \tag{79}$$

Combine Eq. (75) and (79) to obtain $\sum_{t=1}^{T} \boldsymbol{g}_t \cdot (\boldsymbol{\pi}_t - \boldsymbol{\pi}) \leq \frac{1}{\eta} \left( D_{\Phi,\max}^2 + 2L_\Phi P_T \right) + \eta \sum_{t=1}^{T} \Lambda \left( \nabla_{\boldsymbol{\pi}} f_t(\boldsymbol{\pi}_t), \boldsymbol{\pi}_t \right)$. Consider the bound on $\Lambda \left( \nabla_{\boldsymbol{\pi}} f_t(\boldsymbol{\pi}_t), \boldsymbol{\pi}_t \right)$ in Lemma 8. We obtain

$$\sum_{t=1}^{T} \boldsymbol{g}_t \cdot (\boldsymbol{\pi}_t - \boldsymbol{\pi}) \leq \frac{1}{\eta} \left( D_{\Phi,\max}^2 + 2L_\Phi P_T \right) + \eta \sum_{t=1}^{T} \frac{\|\boldsymbol{g}_t\|_\star^2}{2\rho} \leq \frac{1}{\eta} \left( D_{\Phi,\max}^2 + 2L_\Phi P_T \right) + \eta \left( \frac{L^2 T}{2\rho} \right). \tag{80}$$

Moreover, the following holds:

$$\left\| \boldsymbol{\pi}_{t+1} - \boldsymbol{\pi}_t \right\|_1 \leq \sqrt{\frac{2}{\rho} D_\Phi(\boldsymbol{\pi}_t, \boldsymbol{\pi}_{t+1})} \leq \sqrt{\frac{2}{\rho} D_\Phi(\boldsymbol{\pi}_t, \boldsymbol{z}_{t+1}) - \frac{2}{\rho} D_\Phi(\boldsymbol{\pi}_{t+1}, \boldsymbol{z}_{t+1})} \leq \sqrt{\frac{2}{\rho} D_\Phi(\boldsymbol{\pi}_t, \boldsymbol{z}_{t+1})} \tag{81}$$

$$\leq \sqrt{2\eta^2 \frac{L^2}{2\rho^2}} \leq \frac{L\eta}{\rho}. \tag{82}$$

The above chain of inequalities is obtained through: the strong convexity of $\Phi$, the generalized Pythagorean equality (67), non-negativity of the Bregman divergence of a convex function, and Lemma 8, in respective order. Thus, it holds $\sum_{t=1}^{T} \left\| \boldsymbol{\pi}_{t+1} - \boldsymbol{\pi}_t \right\| = \mathcal{O}(\eta T)$. We conclude the proof. $\qquad\square$

### C.3 Entropic OMD Instantiation

**Definition 5.** *The entropic OMD* (57) *is defined for the negative entropy mirror map*

$$\Phi(\boldsymbol{\pi}) : \boldsymbol{\pi} \in \mathbb{R}_{>0}^{\mathcal{X}} \to \sum_{i=1}^{|\mathcal{X}|} \pi_i \log(\pi_i). \tag{83}$$

The Entropic OMD algorithm is an attractive choice for simplex decision sets because its regret bounds exhibit better dependence on the problem dimension than those of OGD (Online Gradient Descent) [23]. Additionally, our restriction of the simplex decision set $\triangle_{\mathcal{X},\sigma} \subseteq \triangle_{\mathcal{X}}$ allows us to extend Entropic OMD to the dynamic regret setting while preserving its aforementioned advantage. Formally,

**Proposition 4.** *The Entropic OMD algorithm initialized with the state $\boldsymbol{\pi}_1 = \frac{1}{|\mathcal{X}|}$ and configured over the decision set $\triangle_{\mathcal{X},\sigma}$ satisfies Assumptions (4), (5), and (6) with the following quantities:*

- *The map $\Phi$ is 1-strongly convex w.r.t. the $\|\cdot\|_1$ over the $|\mathcal{X}|$-dimensional subset of the simplex $\triangle_{\mathcal{X},\sigma}$.*

- *The map $\Phi$ has bounded gradient over $\triangle_{\mathcal{X},\sigma}$ given by $L_\Phi = |\log(1/\sigma) + 1|$.*

- *The Bregman divergence $D_\Phi(\boldsymbol{\pi}, \boldsymbol{\pi}_1)$ associated to $\Phi$ is bounded over $\mathcal{S}$ by $D_{\Phi,\max} = \log(|\mathcal{X}|)$.*

*Proof.* The negative entropy is 1-strongly convex over the simplex, a result that can be induced from Pinker's inequality. The gradient of the mirror map is given by $\nabla\Phi(\boldsymbol{\pi}) = (\log(\boldsymbol{\pi}_{\boldsymbol{x}}) + 1)_{\boldsymbol{x} \in \mathcal{X}}$, which can take at most a value of $L_\Phi = \log(1/\sigma)$ under the $\|\cdot\|_\infty$ over $\triangle_{\mathcal{X},\sigma}$. The Bregman divergence associated to $\Phi$ is given by

$$D_\Phi(\boldsymbol{\pi}, \boldsymbol{\pi}_1) = \Phi(\boldsymbol{\pi}) - \Phi(\boldsymbol{\pi}_1) - \nabla\Phi(\boldsymbol{\pi}_1)(\boldsymbol{\pi} - \boldsymbol{\pi}_1) \leq \Phi(\boldsymbol{\pi}) - \boldsymbol{\pi}(\boldsymbol{\pi}_1) \leq -\Phi(\boldsymbol{\pi}_1) = \log(|\mathcal{X}|).$$

The first inequality is obtained from first order optimality condition combined with the fact that $\boldsymbol{\pi}_1 = \operatorname{argmin}_{\boldsymbol{x} \in \triangle_{\mathcal{X},\sigma}} \Phi(\boldsymbol{\pi})$. The second inequality is obtain considering that the map $\Phi$ is non-positive for every $\boldsymbol{\pi} \in \triangle_{\mathcal{X},\sigma}$. $\qquad\square$

The same proposition evidently holds over the set $\triangle_{\mathcal{T},\sigma'}$ for $\sigma' \in [0, 1]$.

### C.4 Entropic-VBR Policy

The Entropic-VBR policy is designed to operate in two distinct but interrelated settings, as outlined in Section 3. This policy is defined by a pair of mirror descent algorithms, each employing distinct gradients and decision sets. The first decision set, $\triangle_{\mathcal{X}}$, constructs distributions over the experiment space, while the second, $\triangle_{\mathcal{T}}$, generates distributions over the data freshness window space. The policy iteratively updates its decisions, initialized with uniform distributions $\boldsymbol{\pi}_1 = \mathbf{1}/|\mathcal{X}|$ and $\boldsymbol{p}_1 = \mathbf{1}/(\tau+1)$. At time step $t$, the policy is refined according to the following procedure:

$$\pi_{t+1,\boldsymbol{x}} = \Pi_{\triangle_{\mathcal{X},\sigma}}\left(\pi_{\boldsymbol{x}} \exp\left(\eta_{\mathcal{X}}\left(\boldsymbol{x}_t^\mathsf{T}\left(\boldsymbol{V}((\tau+1)\boldsymbol{\pi}_t)\right)^{-1}\boldsymbol{x}\right)^2\right)\right), \qquad \text{for } \boldsymbol{x} \in \mathcal{X} \quad (84)$$

$$p_{t+1,\tau} = \Pi_{\triangle_{\mathcal{T}} \cap [\sigma',1]^{\mathcal{T}}}\left(p_{t,\tau} \exp\left(-\eta_{\mathcal{T}}\frac{\xi_t}{p_\tau}\mathbb{1}\left(\tau = \tau_t\right)\right)\right), \qquad \text{for } \tau \in \mathcal{T},$$

where $\tau_t \sim \boldsymbol{p}_t$ is sampled at every $t$, $\sigma$ is the regularization parameter, and $\sigma'$ is a tuneable parameter.

### C.5 Entropic Variance Reduction Policy

Given the above proposition we obtain the following regret bound for the variance-reduction policy:

**Corollary 2.** *Under Assumptions 1–3, let $\triangle_{\mathcal{X},\sigma}$ be the decision set, and let $\{\boldsymbol{\pi}_t^\star\}_{t=1}^T \in \mathcal{C}(\triangle_{\mathcal{X},\sigma}^T, P_T^{\star,\mathrm{v}})$ be a comparator sequence. Consider the OMD (57) policy with the negative entropy mirror map (83) and learning rate $\eta_{\mathcal{X}} = \Theta\left(\sqrt{\log(1/\sigma)P_T^{\star,\mathrm{v}}T^{-1}}\right)$. If the Entropic OMD is configured to run as a variance reduction policy $\mathcal{L}^v$ against the costs $v_1, \ldots, v_t$, then it has the following regret guarantee:*

$$\mathfrak{R}^v\left(\mathcal{L}^v\right) = \mathcal{O}\left(\sqrt{\log(1/\sigma)P_T^{\star,\mathrm{v}}T}\right). \tag{85}$$

*The policy-induced decisions exhibit a path length (switching cost) of:*

$$\sum_{t=1}^T \|\boldsymbol{\pi}_{t+1} - \boldsymbol{\pi}_t\| = \mathcal{O}\left(\sqrt{P_T^{\star,v}T}\right). \tag{86}$$

This is a direct result from Theorem 3.

### C.6 Entropic Bias Reduction Policy

In this section, we show that the bandit setting can be reduced to the full-information setting via unbiased gradient estimates. In particular, at time $t$ an OMD (57) policy with the negative entropy mirror map (83) and state $\boldsymbol{p}_t$, adapts its state according to the gradient estimate

$$\tilde{\boldsymbol{g}}_t = \frac{\xi_{t,\tau_t}}{p_{\tau_t}}\boldsymbol{e}_{\tau_t}, \tag{87}$$

where $\xi_{t,\tau_t}$ is the feedback for inference window size $\tau_t \in \mathcal{T}$. The estimate satisfies the following:

**Lemma 12.** *The gradient estimate* (87) *satisfies the following for any* $\boldsymbol{p} \in \triangle_{\mathcal{T}}$:

$$\mathbb{E}\left[\tilde{\boldsymbol{g}}_t\right] = \boldsymbol{\xi}_t. \tag{88}$$

*Proof.* Simply, evaluate the expectation: $\mathbb{E}\left[\tilde{\boldsymbol{g}}_t\right] = \sum_{\tau \in \mathcal{T}} p_\tau \frac{\xi_{t,\tau}}{p_\tau} \boldsymbol{e}_\tau = \boldsymbol{\xi}_t.$ ☐

**Theorem 4.** *Under Assumptions 1–3, consider* $\triangle_{\mathcal{T},\sigma'}$ *as the decision set and comparator sequence* $\tau_t^\star \in \mathcal{C}\left(\mathcal{T}^T, P_T^{\mathrm{b}}\right)$ *with path-length* $P_T^{\mathrm{b}}$. *Entropic OMD* (83) *configured to run as a bias-reduction policy* $\mathcal{L}^{\mathrm{b}}$ *for* $\eta_{\mathcal{T}} = \Theta\left(\log(1/\sigma')P_T^{\star,\mathrm{b}}T^{-1/2}\right)$ *and* $\sigma' = \Theta\left(T^{-1/2}\right)$ *against the gradients* $\tilde{\boldsymbol{g}}_1, \ldots, \tilde{\boldsymbol{g}}_T$ (87) *has the following expected regret*

$$\mathfrak{R}^{\mathrm{b}}\left(\mathcal{L}^{\mathrm{b}}\right) = \mathcal{O}\left(\sqrt{\log(T)P_T^{\star,\mathrm{b}}T}\right). \tag{89}$$

*Proof.* Let $\mathcal{H}_t = \{\tau_1, \ldots, \tau_t\}$ be the history of the samples picked by the algorithm. The following holds

$$\mathbb{E}\left[\sum_{t=1}^T \xi_{t,\tau_t} - \sum_{t=1}^T \xi_{\tau_t^\star,t}\right] = \mathbb{E}\left[\sum_{t=1}^T \boldsymbol{\xi}_t \cdot \boldsymbol{p}_t - \sum_{t=1}^T \boldsymbol{\xi}_t \cdot \boldsymbol{p}_t^\star\right] \tag{90}$$

$$= \mathbb{E}\left[\sum_{t=1}^T \boldsymbol{\xi}_t \cdot (\boldsymbol{p}_t - \boldsymbol{p}_t^\star)\right] \tag{91}$$

$$= \mathbb{E}\left[\sum_{t=1}^T \mathbb{E}\left[\tilde{\boldsymbol{g}}_t \cdot (\boldsymbol{p}_t - \boldsymbol{p}_t^\star) \,\middle|\, \mathcal{H}_{t-1}\right]\right]. \tag{92}$$

Consider the following bound on $\mathbb{E}\left[\tilde{\boldsymbol{g}}_t \cdot (\boldsymbol{p}_t - \boldsymbol{p}_t^\star) \,\middle|\, \mathcal{H}_{t-1}\right]$ from Lemma 11

$$\tilde{\boldsymbol{g}}_t \cdot (\boldsymbol{p}_t - \boldsymbol{p}) \leq \frac{1}{\eta}\left(\eta^2 \Lambda(\boldsymbol{g}_t, \boldsymbol{p}_t) + D_\Phi(\boldsymbol{p}, \boldsymbol{p}_t) - D_\Phi(\boldsymbol{p}, \boldsymbol{p}_{t+1})\right) \tag{93}$$

$$\leq \frac{1}{\eta}\left(D_\Phi(\boldsymbol{p}, \boldsymbol{p}_t) - D_\Phi(\boldsymbol{p}, \boldsymbol{p}_{t+1})\right) + \frac{1}{2}\eta \frac{\xi_{t,\tau_t}^2}{p_{t,\tau_t}}. \qquad \text{(Lemma 9)} \tag{94}$$

Evaluate the conditional expectation $\mathbb{E}\left[\cdot \,\middle|\, \mathcal{H}_{t-1}\right]$ to obtain the following bound:

$$\mathbb{E}\left[\frac{1}{2}\eta p_{t,\tau_t}\xi_{t,\tau_t}^2 \,\middle|\, \mathcal{H}_{t-1}\right] = \frac{1}{2}\eta \sum_{\tau \in \mathcal{T}} p_{t,\tau} \frac{\xi_{t,\tau}^2}{p_{t,\tau}} = \frac{\eta}{2}\left\|\boldsymbol{\xi}_t^2\right\|_1. \tag{95}$$

Take $L \geq \max_{t \in [T], \tau \in \mathcal{T}} \xi_{t,\tau}$ (which we know is $\mathcal{O}(1)$ from Asms. 1–3). Note that since the decision set is restricted $\triangle_{\mathcal{T},\sigma'} \subseteq \triangle_{\mathcal{T}}$, so the policy competes with comparator points on $\triangle_{\mathcal{T},\sigma'}$. However, we can project the original comparator points that lie inside $\triangle_{\mathcal{T}}$ to $\triangle_{\mathcal{T},\sigma'}$ with a cost difference of at most $L|\mathcal{X}|\sigma'$, by utilizing the Lipchitzness of the expected cost $\boldsymbol{p} \cdot \boldsymbol{\xi}_t$ with $L \geq \max_t \|\boldsymbol{\xi}_t\|_\infty$. Thus, we have:

Thus, we have:

$$\mathbb{E}\left[\sum_{t=1}^T \xi_{t,\tau_t} - \sum_{t=1}^T \xi_{\tau_t^\star,t}\right] \leq \eta \frac{|\mathcal{X}|L^2}{2}T + \frac{1}{\eta}\mathbb{E}\left[\sum_{t=1}^T \left(D_\Phi(\boldsymbol{p}_t^\star, \boldsymbol{p}_t) - D_\Phi(\boldsymbol{p}_t^\star, \boldsymbol{p}_{t+1})\right)\right] + L|\mathcal{X}|\sigma'T \tag{96}$$

$$\leq \eta \frac{|\mathcal{X}|L^2}{2}T + \frac{1}{\eta}\left(\log^2(|\mathcal{X}|) + 2\log(1/\sigma')\sum_{t \in [T]}\left\|\boldsymbol{p}_{t+1}^\star - \boldsymbol{p}_t^\star\right\|_1\right) + L|\mathcal{X}|\sigma'T \tag{97}$$

$$= \eta \frac{|\mathcal{X}|L^2}{2}T + \frac{1}{\eta}\left(\log^2(|\mathcal{X}|) + 4\log(1/\sigma')P_T^{\mathrm{b}}\right) + L|\mathcal{X}|\sigma'T. \tag{98}$$

Select $\eta$ and $\sigma'$ as stated in the theorem to obtain the desired bound. We conclude the proof. ☐

## D Proof of Theorem 1

*Proof.* The theorem follows directly from Theorem 1, Corollary 2, and Theorem 4. By substituting the respective regret bounds, we obtain the desired result. □

## E Proof of Theorem 1

*Proof.* Lemma 6 is applied twice to bound the difference between the time-varying experimental designs and fixed designs. First, we bound the difference between the designs generated by the policy and fixed designs. Subsequently, we bound the difference between the dynamic optimal design and fixed designs.

$$
\mathfrak{R}_T\left(\mathcal{L}^{\mathrm{v+b}}\right) = \mathbb{E}\left[\sum_{t=\tau+1}^{T} f_t(\boldsymbol{\pi}_{t-\tau_t}, \ldots, \boldsymbol{\pi}_t) - \sum_{t=t=\tau+1}^{T} f_t(\boldsymbol{\pi}^\star_{t-\tau^\star_t}, \ldots, \boldsymbol{\pi}^\star_t)\right]
$$

$$
= \mathbb{E}\left[\sum_{t=\tau+1}^{T} f_t(\boldsymbol{\pi}_{t-\tau_t}, \ldots, \boldsymbol{\pi}_t) - \sum_{t=\tau+1}^{T} f_t(\underbrace{\boldsymbol{\pi}^\star_t, \ldots, \boldsymbol{\pi}^\star_t}_{\tau^\star_t})\right] + \sum_{t=1}^{T} \epsilon_{\boldsymbol{\lambda}^\star,t,\tau}
$$

$$
= \mathbb{E}\left[\sum_{t=\tau+1}^{T} f_t(\boldsymbol{\pi}_{t-\tau_t}, \ldots, \boldsymbol{\pi}_t) - \sum_{t=\tau+1}^{T} f_t(\boldsymbol{\pi}_{t-\tau^\star_t}, \ldots, \boldsymbol{\pi}_t)\right] +
$$

$$
\mathbb{E}\left[\sum_{t=\tau+1}^{T} f_t(\boldsymbol{\pi}_{t-\tau^\star_t}, \ldots, \boldsymbol{\pi}_t) - \sum_{t=\tau+1}^{T} f_t(\underbrace{\boldsymbol{\pi}^\star_t, \ldots, \boldsymbol{\pi}^\star_t}_{\tau^\star_t})\right] + \sum_{t=1}^{T} \epsilon_{\boldsymbol{\lambda}^\star,t,\tau}
$$

$$
\leq \mathfrak{R}^{\mathrm{b}}_T\left(\mathcal{L}^{\mathrm{b}}\right) + \mathbb{E}\left[\sum_{t=\tau+1}^{T} f_t(\underbrace{\boldsymbol{\pi}_t, \ldots, \boldsymbol{\pi}_t}_{\tau^\star_t}) - \sum_{t=\tau+1}^{T} f_t(\underbrace{\boldsymbol{\pi}^\star_t, \ldots, \boldsymbol{\pi}^\star_t}_{\tau^\star_t})\right] + \sum_{t=\tau+1}^{T} \epsilon_{\boldsymbol{\lambda}^\star,t,\tau} + \sum_{t=\tau+1}^{T} \epsilon_{\boldsymbol{\lambda},t,\tau}
$$

$$
= \mathfrak{R}^{\mathrm{b}}_T\left(\mathcal{L}^{\mathrm{b}}\right) + \mathbb{E}\left[\sum_{t=\tau+1}^{T} f_t(\underbrace{\boldsymbol{\pi}_t, \ldots, \boldsymbol{\pi}_t}_{\tau^\star_t}) - \sum_{t=\tau+1}^{T} f_t(\underbrace{\boldsymbol{\pi}^\star_t, \ldots, \boldsymbol{\pi}^\star_t}_{\tau^\star_t})\right] + \sum_{t=\tau+1}^{T} \epsilon_{\boldsymbol{\lambda}^\star,t,\tau} + \sum_{t=\tau+1}^{T} \epsilon_{\boldsymbol{\lambda},t,\tau}
$$

$$
\leq \mathfrak{R}^{\mathrm{b}}_T\left(\mathcal{L}^{\mathrm{b}}\right) + \mathfrak{R}^{\mathrm{v}}_T\left(\mathcal{L}^{\mathrm{v}}\right) + \sum_{t=\tau+1}^{T} \epsilon_{\boldsymbol{\lambda}^\star,t,\tau} + \sum_{t=\tau+1}^{T} \epsilon_{\boldsymbol{\lambda},t,\tau}
$$

$$
= \mathfrak{R}^{\mathrm{b}}_T\left(\mathcal{L}^{\mathrm{b}}\right) + \mathfrak{R}^{\mathrm{v}}_T\left(\mathcal{L}^{\mathrm{v}}\right) + \mathcal{O}\left(P^v_T + P^{\star,v}_T\right) \quad \text{(Proposition 3).} \tag{99}
$$

The final inequality follows from the observation that the EPE of a fixed design has a constant bias term from Lemma 5, while the variance term is scaled by $\frac{1}{\tau_t+1}$, which is always less than or equal to 1. We conclude the proof. □

## F Additional Details on Experimental Setup

We consider the data domain, denoted by $\mathcal{X}$, as a subset of an ellipse centered at the origin. Within this region, we generated a uniform grid of points, as illustrated in Figure 2 (a) (points covered by the algorithm's initial state). We considered a time horizon of $T = 3 \times 10^4$ query points. These query points, $\boldsymbol{x}_t$, were sampled from a truncated Gaussian distribution characterized by a standard deviation of $\sigma = 1.5$. The distribution's center was shifted to $(5, 0)$, $(-5, -5)$, and subsequently back to $(5, 0)$ at times $\frac{1}{3}T$ and $\frac{2}{3}T$, respectively, to simulate a dynamic query distribution pattern. Figure 2 (c) provides a visual representation of the evolving query distribution. The true underlying model, $\boldsymbol{\beta}^\star_1$, was initially set to $(2, 2)$. To introduce temporal dynamics, we modeled its evolution as a random walk

$$
\boldsymbol{\beta}^\star_{t+1} = \Pi_{[0,5]^2}\left(\boldsymbol{\beta}^\star_t + 0.3\eta_t\right), \tag{100}
$$

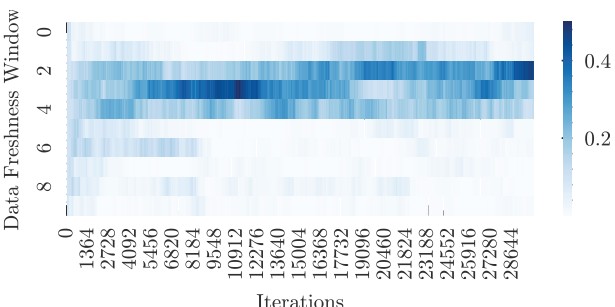

Figure 3: Evolution of the policy-learned distribution over time, illustrating the non-trivial nature of the optimal window size.

where $\Pi_{[0,5]^2}$ is a projection operator that ensures the model parameters remain within the interval $[0,5]$, $\eta_t$ is a Rademacher random variable taking values from $\{-1, 1\}$. The evolution of the model parameters is depicted in Figure 2 (a). For our experiments, we set the experimental budget to $M = 100$, the data-freshness window size to $\tau = 10$, and the noise of the labels to a standard Gaussian scaled by 16.

