# OpenReview forum: "Adaptive Transductive Inference via Sequential Experimental Design with Contextual Retention"
_NeurIPS.cc/2024/Workshop/BDU — NeurIPS BDU Workshop 2024 Poster_

### Official Review · Reviewer_J3eF · 2024-09-22
**The paper presents a three-stage framework for active learning, which includes data collection, model retraining, and deployment, aiming to optimize data acquisition, data freshness, and model selection methodologies in dynamic environments. The proposed approach integrates principles from sequential optimal experimental design and online learning. Some theoretical analysis and empirical evaluations demonstrate the efficacy of the proposed method.**

**Rating:** 7
**Confidence:** 2

**Review:**

Quality:

Pros:
- The paper provides a solid theoretical mathematical foundation with many definitions, proof and lemma.
- Empirical results are presented to show improvements over the baselines.

Cons:
- The experiments are conducted in synthetic settings, which may not reflect the the distribution of the real-data.
- Some assumptions made in the theoretical analysis might limit the practical applicability in real-world scenarios.


Clarity:

Pros:
- The paper is generally well-organized, with a clear flow from problem formulation to solution and experimental results.

Cons:
- The theoretical theories and advanced concepts can benefit from some context information to improve the readablity.
- Some demonstrations are not explicit, e.g., in the figure (a), (b), what's the meaning of x-axis and y-axis, how the distributions/designed are changed?

Originality:

Pros:
- The introduction of a three-stage framework that integrates sequential experimental design looks promising.
- The concept of balancing data freshness with experimental design in an online setting is innovative.

Significance:

Pros:
- The proposed framework is highly relevant for applications involving large-scale data and resource constraints, and provides significant guidances on optimizing data acquisition and model updates.

Cons:
- The reliance on specific assumptions (e.g., data retention policy) might restrict the applicability and lack of performance comparision on real-world datasets beyond the synthetic experiments presented.

---

### Official Review · Reviewer_PXsm · 2024-09-26
**A very involved "frequentist" approach to experimental design**

**Rating:** 3
**Confidence:** 2

**Review:**

### Summary
The paper proposes a three-level framework for experimental design with performance guarantees. These three steps consist of data collection, model retraining and model deployment, which are highly relevant to real-life ML practitioners' concerns.

### Strengths
- The theoretical foundation, performance guarantees and exhaustive proofs make it a solid study.

### Weaknesses
- I could not spot any Bayesian approach in the paper. Concepts like "bias-variance tradeoff" and "risk minimization" belong mostly to frequentist statistics. Therefore, I doubt whether this study is appropriate for this workshop.
- It feels like it is part (or continuation) of a bigger study, which is fine. However, the preliminaries for experimental design are highly missing. For example, "a continuous fraction in [0,1] of these experiments is allocated to $\mathcal{X}$" is a vague statement for those who do not come from the experimental design field. Please either be more descriptive (also for the notation $[0, 1]^\mathcal{X})$ or give on-point references.
- Again, due to the lack of references and background material, it is hard to assess where the novelty starts in the proposed framework.
- Numerical experiments are barely understandable from Figure 2.
- A conclusion that wraps up everything is missing.

---

### Decision · Program_Chairs · 2024-10-09

**Decision:**

Accept (Poster)

**Comment:**

Reviews for this paper are mixed. The negative review is essentially worried about workshop fit. However, the call for papers explicitly mentions "sequential experimental design" as a topic, and though the workshop explicitly encourages Bayesian approaches, we do not have a policy that excludes other approaches that the authors thought were sufficiently relevant to our scope to submit them in the first place. Therefore, I recommend following the other reviewer, and accepting the work.